# VRL3: A Data-Driven Framework for Visual Deep Reinforcement Learning

**Che Wang**[1,2*]      **Xufang Luo**[3]      **Keith Ross**[1]      **Dongsheng Li**[3]

[1] New York University Shanghai
[2] New York University
[3] Microsoft Research Asia, Shanghai, China

## Abstract

We propose VRL3, a powerful data-driven framework with a simple design for solving challenging visual deep reinforcement learning (DRL) tasks. We analyze a number of major obstacles in taking a data-driven approach, and present a suite of design principles, novel findings, and critical insights about data-driven visual DRL. Our framework has three stages: in stage 1, we leverage non-RL datasets (e.g. ImageNet) to learn task-agnostic visual representations; in stage 2, we use offline RL data (e.g. a limited number of expert demonstrations) to convert the task-agnostic representations into more powerful task-specific representations; in stage 3, we fine-tune the agent with online RL. On a set of challenging hand manipulation tasks with sparse reward and realistic visual inputs, compared to the previous SOTA, VRL3 achieves an average of 780% better sample efficiency. And on the hardest task, VRL3 is 1220% more sample efficient (2440% when using a wider encoder) and solves the task with only 10% of the computation. These significant results clearly demonstrate the great potential of data-driven deep reinforcement learning.

## 1 Introduction

Over the past few years, the sample efficiency of Deep Reinforcement Learning (DRL) has significantly improved in popular benchmarks such as Gym [4, 82] and DeepMind Control Suite (DMC) [80]. However, the environments in these benchmarks often look different from the real world (see discussion in section 5.8, page 8 in [73]). It remains unclear whether the sample-efficient methodologies developed for these benchmarks can be successfully extended to more practical tasks that rely on realistic visual inputs from cameras and sensors, such as in robotic control and autonomous driving.

A promising direction is to take a data-driven approach, that is, try to use all the data available that might contribute to the learning process. Indeed, in the past few decades, the most important advances in deep learning have been facilitated by the use of large amounts of data and computation [78, 47]. Such a data-driven approach is natural and mature for supervised learning, but not as much so for RL, where learning is most commonly performed in an online fashion.

Although recent advances in representation learning for RL and offline RL reveal the potential of such an approach, most existing offline RL methods focus on tasks with proprioceptive ("raw-state") input and not the challenging visual control tasks. There is yet to be a method that can fully utilize data from multiple different sources to boost learning efficiency. In this work, the goal is to develop a data-driven framework that fills this role. And this framework is designed to be as simple as

---

*This work was done when Che Wang was interning with Microsoft Research Asia.

36th Conference on Neural Information Processing Systems (NeurIPS 2022).

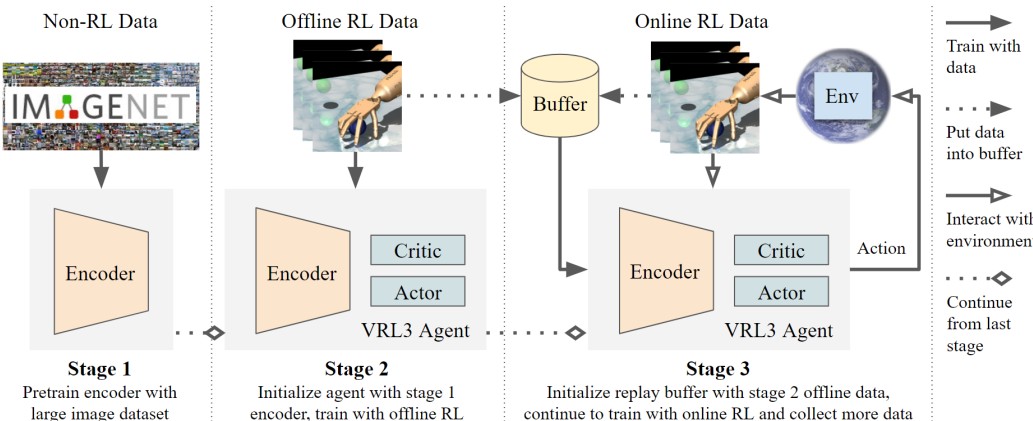

Figure 1: The design of VRL3. In stage 1, we pretrain an encoder using large image datasets to obtain task-agnostic visual representation. In stage 2, we initialize an actor-critic RL agent with the pretrained encoder, and then use offline RL techniques to train on offline RL data. Here we finetune the encoder and also learn the actor and critic. In stage 3, we initialize a buffer with the offline data from stage 2, and further train the entire agent with online RL. The idea is simple: to address challenging visual control tasks, the goal is to fully exploit all the available data.

possible while being sample efficient. As discussed in a number of critique papers [27, 28, 17], simplicity in algorithm design is preferred because the performance of RL algorithms can be heavily affected by hidden details in the code, and a simpler design often leads to cleaner analysis and better reproducibility. Note that VRL3 is clearly not simpler when compared to a purely online method that only trains in 1 stage and does not take advantage of data from other sources. However, VRL3 is simpler than other alternative ways to utilize data from 3 separate stages. In the later sections, we provide extensive discussions and ablations to show that VRL3 indeed has a simple and intuitive design that can fully exploit all data sources to reach superior performance.

We also make an effort to maximize robustness and reproducibility, by comparing to stronger baselines that we re-implemented and improved, presenting extensive ablations and a hyperparameter robustness study, listing full technical details, providing open source code, and more. (Source code is being reviewed and cleaned and will be put on Github soon).

Figure 1 shows the proposed framework, Visual Deep Reinforcement Learning in 3 Stages (VRL3). VRL3 is designed with a data-driven mindset, and each stage utilizes a particular type of data source.

In stage 1, we learn from large, existing non-RL datasets such as the ImageNet dataset. These datasets might not be directly relevant to the RL task, but they provide task-agnostic visual representations that might be helpful, especially if the RL task also has realistic visuals. A major challenge here is that the non-RL data can come from a different domain with different input shapes. Special care is required to make sure the representations obtained here can be smoothly transferred to the RL task.

In stage 2, we learn from offline RL data, such as a small number of expert demonstrations, or data collected by a previous learning agent. This offline data is related to the RL task and contains important information about solving the task. We initialize an actor-critic RL agent with the pretrained encoder and then employ offline RL techniques which not only train the actor-critic RL agent but also fine-tune the encoder, thereby creating a task-specific representation.

In stage 3, we learn from online RL data. Our main goal is sample efficiency (achieve high performance with as little online data as possible). We, therefore, use off-policy training, which uses and re-uses all available data during training. Note that naively transitioning into standard online RL can lead to training instability and destroy what we learned from the previous stages. We avoid this instability by constraining the maximum Q target value during off-policy updates.

Although each stage brings a number of unique challenges, we show that VRL3 can address all of them and take full advantage of the different data sources. Our contributions are as follows:

1. Novel framework: we present a data-driven framework that can fully exploit non-RL, offline RL, and online RL data to solve challenging tasks with sparse reward and realistic visual input. Most existing visual methods do not utilize offline data, and most offline RL methods do not work with difficult visual tasks, making our framework a novel contribution. We provide source code [2] and a full set of technical details to maximize reproducibility.

2. Novel technical contributions: we propose convolutional channel expansion (CCE) to enable an ImageNet-pretrained encoder to work with RL tasks that take in multiple input images. We also propose the Safe Q target technique for minimal-effort offline to online transition.

3. Novel insights: we show that offline RL data (expert demonstrations) can provide a benefit that cannot be replaced by other stages, and that conservative RL might be a superior option when exploiting offline data, compared to contrastive learning. We also show that a reduced encoder learning rate helps learning in a multi-stage setting. And interestingly, representation pretrained with non-RL data, despite the large domain gap, can give better performance compared to representation learned from limited RL data, but is outperformed when RL data is ample. We also find stage 1 pretraining helps more in more difficult tasks.

4. Novel results: We present extensive ablations that reveal the effect of each component of VRL3, and conduct a large hyperparameter study on tasks with different difficulties, providing useful guidelines on how to deploy and tune our framework to new tasks.

5. Significant improvement over the previous SOTA: VRL3 achieves a new level of SOTA sample efficiency (780% better on average), parameter efficiency (3 times better), and computation efficiency (10 times faster to solve hardest task) on the challenging Adroit benchmark while also being competitive on DMC. When using a wider encoder, we can even reach 2440% better sample efficiency on the hardest Relocate task compared to the previous SOTA.

## 2 Related Work

VRL3 has a straightforward design and effectively combines representation learning, offline RL, and online RL. We now discuss related work and how our contribution is novel and different from prior works, with a focus on model-free methods. Additional related work is discussed in Appendix D.

### 2.1 Learning visual tasks with online RL

With the help of data augmentation, many visual RL tasks can be learned in an online fashion. Typically, agents trained with such methods use convolutional encoders to extract lower-dimensional features from the visual input, and train in an end-to-end fashion [35, 93, 43, 94]. Another class of methods uses contrastive representation learning to solve visual tasks [44, 77, 99]. Contrastive learning can be used to learn useful representations without gradients from the RL objectives. These methods work well in environments with simple visuals, such as in the DMC environments, but can fail entirely if naively applied to challenging control tasks with sparse rewards and more realistic visuals such as Adroit, due to the difficulty in both exploration [66] and learning representations for realistic visual inputs [73]. Different from these works, VRL3 can fully exploit offline RL data and even non-RL data and can achieve SOTA performance in both Adroit and DMC.

### 2.2 RL with ImageNet pretraining

A number of previous works have used a pretrained encoder, such as a ResNet trained on ImageNet to help convert visual input into low-dimensional features, the encoder can be further finetuned or kept frozen during training [73, 16, 31, 22, 48, 64, 72]. However, such pretraining alone might not be sufficient to achieve the best performance or computation efficiency. Naively initializing the agent with an ImageNet-pretrained encoder might not provide a significant advantage over training from scratch [31]. Most of these works do not incorporate offline RL data. RRL [73] uses offline RL data but does not exploit them to the full extent. Different from these works, we show how ImageNet pretraining can be best combined with offline and online RL and provide a series of novel analyses, interesting results, and important insights. Other pretraining settings are also explored, [90] show

---

[2]https://sites.google.com/nyu.edu/vrl3

masked image pretraining can lead to strong performance on a new suite of robotic tasks. [62] and [59] show that pretraining on different datasets and with more sophisticated methods can be combined with imitation learning. Compared to them, we study RL, which is a different subject, and we also achieve better success rate on Adroit (95%, while they have 85% and <70%, respectively).

## 2.3 Offline RL and data-driven RL

In the past few decades, some of the most important advances in AI research have been based on data-driven methods, so it is natural to ask whether we can also utilize a similar mindset for DRL [47, 78]. Although off-policy algorithms do not naively work in offline settings [18, 49], many specialized algorithms are proposed to allow effective offline training [18, 37, 8, 63, 88, 36, 17, 87, 34, 92, 2, 20, 89]. Recent works also study the transition from offline to online [58, 46, 102, 46, 34, 49]. However, most of these works are tested on tasks with proprioceptive (positions & velocities) input, and they do not consider pretraining from non-RL data. Our work is different and is focused on challenging control tasks with visual input, sparse reward, and limited demonstrations.

Note that Adroit has often been studied with proprioceptive and not the more challenging visual input. RRL [73] and FERM [98] are the most relevant to our work. RRL uses an ImageNet-pretrained encoder, performs behavioral cloning (BC) on offline RL data, and then performs on-policy learning. Prior to our work, RRL is the SOTA on visual Adroit and is the only method that achieves non-trivial performance on the hardest Relocate task. FERM performs contrastive learning on offline RL data and then performs off-policy online RL with data augmentation. FERM can learn simple robotic tasks quickly but has brittle performance in Adroit [73].

In most prior works, the different training stages have been **studied separately**. What is missing is a framework that can **combine all 3 stages effectively and efficiently**. A major novel contribution of this work is such a framework and a comprehensive study on how its components interact and build up towards superior performance.

## 3 VRL3: Visual DRL in 3 stages

We now present the design of our framework VRL3. We focus on the high-level ideas and provide additional technical details in Appendix A and pseudocode in appendix C. Our agent has a convolutional encoder, two Q networks, two target Q networks and a policy network. Note the encoder can be trained separately from the policy and Q networks.

**Stage 1**  We follow the standard supervised learning routine and train a convolutional encoder $f_\xi$ on a 1000-class ImageNet classification task. We follow the standard training procedure of a typical ResNet model. During training, we shrink the images to the size of 84x84 so that they match the input dimension of our downstream RL task. In our main results, we also use a lightweight encoder with 5 convolutional layers, with batch norm after each layer. In appendix A.7 we show additional results with deeper ResNet models.

***Transition to multi-frame RL input***  After stage 1, we expand the first convolutional layer of the encoder so the input channel size matches the RL task which can take in multiple frames, and the weight values in this layer are divided accordingly to maintain features to later layers, we refer to this as **convolutional channel expansion (CCE)** (Further discussed in Section 5).

**Stage 2**  We first initialize an RL agent with the pretrained encoder and put the offline RL data in an empty replay buffer $\mathcal{D}$. Note that for Adroit, we have a standard 25 expert demonstrations per task (collected by human users with VR) [66]. We build our implementation on top of DrQv2 because it is a simple off-policy actor-critic algorithm with SOTA performance and a clean and efficient codebase [93]. Here we also try to follow the notations in DrQv2 for better consistency.

***Input and image augmentation***  Let aug denote random shift image augmentation, and cat for concatenation. Let $x$ be the visual input from the camera (or cameras). In the Adroit environment, in addition to the visual observation, we are also given sensor values from the robotic hand, which is a vector of real numbers, denoted by $z$. The augmented visual input (image frames) is put into the encoder, giving a lower-dimensional representation $h = f_\xi(\text{aug}(x))$. We then concatenate it with the

sensor values (this is skipped in DMC). For simplicity, let $s = \text{cat}(h, z)$ denote the concatenation; $s$ is the input to the Q and policy networks, which are multi-layer perceptrons.

***Offline RL update*** For each update, we sample a mini-batch of transitions $(x_t, z_t, a_t, r_{t:t+n-1}, x_{t+n}, z_{t+n})$ from the replay buffer $\mathcal{D}$. Note we have $t + n$ here to compute n-step returns [93]. Let $s_t = \text{cat}(h_t, z_t)$. Let $\tilde{a}_t \sim \pi_\phi(s_t)$ be actions sampled from the current policy. The Q target value $y$ is: $y = \sum_{i=0}^{n-1} \gamma^i r_{t+i} + \gamma^n \min_{k=1,2} Q_{\bar{\theta}_k}(s_t, \tilde{a}_{t+n})$.

The standard Q loss $\mathcal{L}_Q$ is (for $k = 1, 2$): $\mathcal{L}_Q(\mathcal{D}) = \mathbb{E}_{\tau \sim \mathcal{D}}\big[(Q_{\theta_k}(h_t, a_t) - y)^2\big]$.

Following [37], and build on top of our framework, we use an additional conservative Q loss $\mathcal{L}_{QC}$ term: $\mathcal{L}_{QC}(\mathcal{D}) = \mathbb{E}_{\tau \sim \mathcal{D}}\big[\log \sum_{\tilde{a}_t} \exp(Q(s_t, \tilde{a}_t)) - Q(s_t, a_t)\big]$. On a high-level, this loss essentially reduces the Q value for actions proposed by the current policy, and increase the Q values for actions in the replay buffer [37].

The standard policy (actor) loss $\mathcal{L}_\pi$ is: $\mathcal{L}_\pi(\mathcal{D}) = -\mathbb{E}_{\tau \sim \mathcal{D}}\big[\min_{k=1,2} Q_{\theta_k}(s_t, a_t)\big]$.

Let $\alpha$ be the learning rate for the policy network and the Q networks. Let $\beta_{\text{enc}}$ be the encoder learning rate scale, so that the encoder learning rate is $\alpha_{\text{enc}} = \beta_{\text{enc}}\alpha$. For each offline RL update in stage 2, we perform gradient descent on $\mathcal{L}_Q(\mathcal{D}) + \mathcal{L}_{QC}(\mathcal{D})$ to update the encoder $f_\xi$ and the Q networks $Q_{\theta_1}, Q_{\theta_2}$, and gradient descent on $\mathcal{L}_\pi$ to update the policy $\pi_\phi$. Note that we do not use the gradient from the policy to update the encoder, following previous work [94, 93]. We also use Polyak averaging hyperparameter $\tau$ to update target networks, as is typically done.

***Safe Q target technique*** When computing the Q target in stages 2 and 3, we propose a simple yet effective solution to prevent potential Q divergence due to training instability and distribution shift. For all tasks, we set a maximum Q target value. To decide this threshold value, we reshape the maximum per-step reward $r_{\max}$ to be 1 and estimate what the maximum Q value should be for an optimal policy (Details in Appendix A.2). To be concrete, we can compute the sum of this infinite geometric series, $Q_{\max} = \sum_{t=0}^{\infty} \gamma^t r_{\max} = 100$ when $\gamma = 0.99$ ($\gamma$ is discount factor). Empirical results show that this computed threshold is robust and does not require tuning (as shown in Figure 1e in the appendix A, slightly lower or higher threshold values such as 50 and 200 also give similar performance). Now if during any Q update, a Q target value $y$ exceeds this threshold value, then we reduce it to be closer to the threshold value: if $y > Q_{max} + 1$, then $y \leftarrow Q_{\max} + (y - Q_{\max})^\eta$, where $0 \leq \eta \leq 1$ is the safe Q factor. Notice the $Q_{\max} + 1$ here is a technical detail to address edge cases.

***Reduced encoder learning rate*** Due to the domain gap between ImageNet and the RL task, we apply a reduced encoder learning rate to avoid the disturbance of pretrained features in stages 2 and 3.

**Stage 3** We continue to use the replay buffer $\mathcal{D}$, which already contains the offline RL data. We now perform standard online RL, and newly collected data are added to $\mathcal{D}$. The update is the same as in stage 2, except we remove the conservative Q loss $\mathcal{L}_{QC}(D)$.

# 4    Results

Figure 2a shows success rate comparison averaged over all 4 Adroit tasks, Figure 2b shows results on the most challenging Relocate task, and Figure 2c shows average performance comparison on all 24 DMC tasks. We use the following methods as our baselines. DrQv2 is an off-policy online RL method that has the SOTA performance on DMC. RRL is an on-policy method that uses pretrained ImageNet features and utilizes demonstrations with behavioral cloning and has SOTA performance on Adroit. FERM is a 2-stage method that first trains an encoder with contrastive learning on offline RL data and then perform online RL. Note that currently there are not many successful multi-stage visual RL frameworks like VRL3, and as discussed in section 2, most existing offline-online methods are not designed to work with the more challenging visual inputs, so they are not included.

**Getting stronger baselines** Naively, DrQv2 does not use offline data and does not work well in Adroit due to the difficulty in exploration. And FERM has brittle performance on Adroit [73]. To create stronger baselines, we introduce variants of DrQv2 and FERM, which are both built on top of our framework. Specifically, DrQv2fD(VRL3) does not have stage 2 training but starts stage 3 training with demonstrations in the buffer. FERM(VRL3) is our re-implementation based on VRL3 and has much stronger performance compared to the results in [73] (more robust, is about 2 times

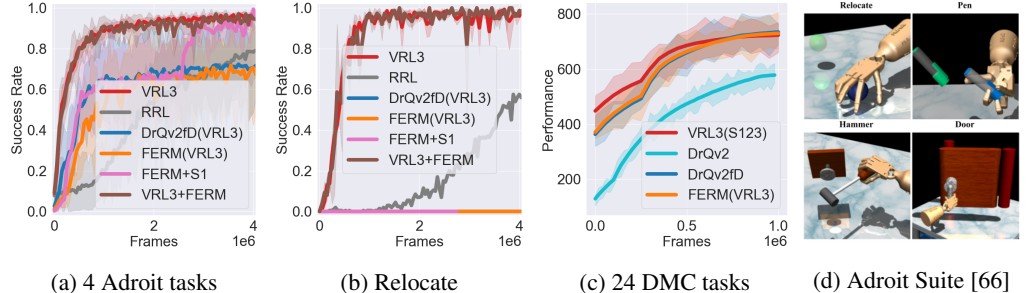

| (a) 4 Adroit tasks | (b) Relocate | (c) 24 DMC tasks | (d) Adroit Suite [66] |

Figure 2: (a) Average success rate over 4 visual Adroit tasks. The performance for VRL3 is averaged over 10 seeds (seeds 0-9). To ensure consistency with prior work, results for RRL (Adroit SOTA) are provided by the authors. VRL3 achieves the strongest performance with a simple design. (b) Success rate for the hardest Relocate task. (c) Average performance over all 24 DMC tasks. VRL3 is still the best, though the performance gap is smaller. Results for DrQv2 are provided by the authors.

faster, and reaches a 1.25x average success rate). For Adroit, we also compare with FERM+S1, which is FERM(VRL3) plus S1 pretraining, and VRL3+FERM, which is VRL3 with additional contrastive encoder learning in stage 2. For these improved baselines, we use the hyperparameters we searched for VRL3 for maximum performance. Doing so helps to ensure our comparison actually reflects the differences in algorithmic design and not hidden implementation details [17, 27].

Figure 2a shows that interestingly, for Adroit, both FERM(VRL3) and DrQv2fD(VRL3) outperform RRL in the early stage, showing that with the right implementation, these off-policy agents can learn quickly even without stage 1 representations. Figure 2b shows that on the most challenging Relocate task, DrQv2fD and FERM variants fail to learn entirely because they do not exploit stage 2 data sufficiently. VRL3 greatly outperforms all other methods.

**Significantly stronger SOTA sample efficiency on Adroit:** Compared to RRL, VRL3 achieves an average of 780% better sample efficiency over all tasks and is 1220% faster (2440% when finetuned) to solve the most difficult Relocate task (in terms of the data required to reach 90% success rate). VRL3+FERM has the same performance as VRL3, showing that additional contrastive learning is not necessary. FERM+S1 is stronger than FERM, showing that stage 1 pretraining also works well with contrastive stage 2 learning. For parameter and computation efficiency, we also use a much smaller encoder compared to RRL. We are 50 times more parameter efficient in the encoder and 3 times more parameter efficient for all networks in the agent. VRL3 is also more computationally efficient, solving the hardest task with just $10\%$ of the computation (Detailed comparison in Appendix B).

For DMC, we collect 25K data with fully trained DrQv2 agents to enable stage 2 training. As shown in Figure 2c, VRL3 has the best performance in the early stage, FERM(VRL3) and DrQv2fD are slightly weaker, but achieve the same performance at 1M data. All three of them outperform DrQv2 (results from authors). Note that DMC is typically treated as an online RL benchmark, and DrQv2 is an online method, so we do not claim a new SOTA. These results are only to show that VRL3 can work in this popular benchmark and can utilize offline data to learn faster. Compared to DMC, Adroit is more challenging due to its sparse reward setting and more complex, more realistic visuals with more depth, lighting, and texture information. This might explain the larger performance gap in Adroit. This shows the advantage of VRL3 is more evident in more challenging visual tasks.

## 5 Challenges, Design Decisions, and Insights

In addition to the strong performance on Adroit, another advantage of VRL3 is its simplicity. The simple design of VRL3 not only makes it much faster and more lightweight than the previous SOTA but also allows us to easily ablate and study its core components. In this section, we discuss some technical challenges and provide a comprehensive analysis of how the different components of VRL3 work together towards superior performance. We focus on Adroit and all experiment results are averaged over 4 Adroit tasks (per-task plots in Appendix A).

## 5.1 Stage 1: Pretraining with non-RL data

The first challenge we face in stage 1 is input shape mismatch. In stage 1, the input to the encoder is a single image with RGB channels. For the downstream RL task, however, it can be beneficial for the agent to take in multiple consecutive frames of images as input, or frames from multiple different cameras (sometimes even a combination of both), so that the agent can learn to use temporal and spatial information in these frames for better performance.

A naive solution is to put each frame separately into the pretrained encoder, as is done in [73]. This naive approach has two downsides: 1) it requires more computation, and 2) even if we finetune the encoder, it cannot learn important temporal and spatial features from the interaction of frames.

**Advantage of convolutional channel expansion (CCE)**: We propose the CCE technique to address the above issue. Let $m$ be the number of input frames for the RL task, With CCE, we expand the first convolutional layer of the stage 1 encoder so that it can now take in inputs of $3m$ channels instead of 3. We repeat the weight matrix $m$ times and then scale all the weight values to $1/m$ of their original values. This allows us to modify the input shape without disrupting the learned representations. Such a simple method (can be done in one line of code, with no additional computation overhead, ablation study against 4 other variants in Appendix A.9) can effectively solve the mismatch problem when transitioning a visual encoder from the non-RL pretraining stage to the RL stages.

To enable faster experimentation and avoid difficulties in training deep networks in DRL [3, 61], we use a lightweight encoder with 5 convolutional layers, which is 50 times smaller than the ResNet34 used in RRL. Quite surprisingly, VRL3 can achieve superior performance with such a tiny encoder.

## 5.2 Stage 2 and Stage 3

**In stage 2, why use conservative RL updates instead of BC or contrastive representation learning?** We use conservative RL for the following reasons: a) we can finetune the encoder to get task-specific representations with RL signal b) comparing with contrastive learning methods which only train the encoder, we can train the encoder as well as the policy and Q networks, giving a jump-start in stage 3. c) With the safe Q technique, this also brings us a minimal-effort design that allows us to transition effectively from stage 2 to stage 3. We can simply remove the conservative term without any elaborate modifications. d) this fits well with an off-policy backbone, allowing maximum data reuse. (BC such as in [66] only works with on-policy methods). Next, we discuss extensive empirical evidence that supports our design.

Figure 3a shows that for VRL3, only using BC on expert demonstrations in stage 2 does not give a good performance. In fact, using BC in stage 2 gives a similar performance to when stage 2 is entirely disabled. Figure 3a shows that for VRL3, only using BC on expert demonstrations in stage 2 leads to the same performance compared to when stage 2 is entirely disabled. Additional ablation can be found in Appendix A.8. This problem is caused by a distribution mismatch of actor and critic at the beginning of stage 3 since BC only trains the actor and not the critic. For contrastive learning, we observe from Figure 2 that FERM has slightly worse performance in DMC and much worse performance in Adroit. This is because Adroit has a more challenging sparse reward setting and FERM fails to fully exploit offline RL data by only updating the encoder. These results show that when tackling challenging tasks, **learning only the representation in stage 2 can be useful, but is not enough to fully exploit offline RL data**. This is a novel and important finding that has never been carefully studied or emphasized in the literature.

**Effect of safe Q target for stage 2-3 transition:** Special care is required to effectively perform off-policy learning in stage 2. In VRL3, we perform conservative updates in stage 2 and remove the conservative term in stage 3. When we transition from stage 2 to stage 3, the new online data will cause a distribution shift that can lead to instability in Q values [58], The safe Q target technique is a simple and efficient solution to this issue. Figure 3b and 3c show the effect of these design decisions: updating with naive RL in stage 2 (S2 Naive) or removing the safe Q technique (No safe Q) lead to severe overestimation in stage 3 and cause performance drop. Updating with conservative RL in stage 3 (S3 Cons) leads to severe underestimation and even worse performance. Naive RL in stage 2 with safe Q in stages 2 and 3 (S2 Naive Safe) can mitigate overestimation but is still weaker than VRL3, which takes conservative updates in stage 2 and use safe Q in stages 2 and 3.

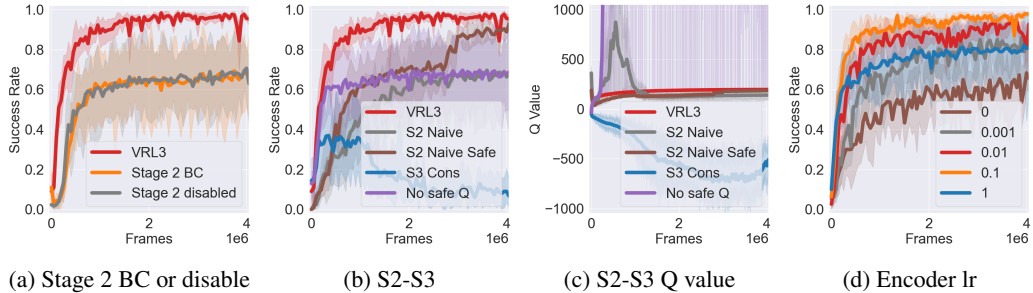

| (a) Stage 2 BC or disable | (b) S2-S3 | (c) S2-S3 Q value | (d) Encoder lr |

Figure 3: a) Effect of BC and offline RL updates in stage 2. Only applying BC or disabling stage 2 training entirely leads to poor performance. b) and c) Taking naive RL updates in stage 2 (S2 Naive) leads to severe overestimation, which can be mitigated by safe Q (S2 Naive Safe). Taking conservative updates in stage 3 (S3 Cons) leads to underestimation. VRL3 gives the best performance with conservative update only in stage 2 and safe Q. Note in our setting, the maximum reasonable Q value should be around 100. d) Effect of different encoder learning rates (e.g., 0.01 means the encoder's learning rate is 100 times slower than that for the policy and Q networks.)

**Reduced encoder learning rate for better finetuning:** When we fine-tune the encoder in stages 2 and 3, a reduced encoder learning rate can prevent pretrained representations from getting destroyed by noisy RL signals, especially in the early stage of RL training. This problem has also been observed in other environments [69]. Figure 3d shows the effect of encoder learning rates. These results show that even though there is a large domain gap between ImageNet and RL tasks, a frozen stage 1 encoder can still provide useful features, and finetuning the encoder can further improve performance. Although it is not new that transfer + finetuning can give the best result [95], it has not been adequately studied in the RL setting with large domain gap and noisy signals.

### 5.3 Contribution of each stage

VRL3 achieves strong performance by fully exploiting data from 3 different stages. We now try to understand how much each stage contributes to the final performance. We perform 2 sets of comparisons, for each VRL3 variant in Figure 4a, we disable one or more stages of training entirely (e.g. S13 means stage 2 training is removed), and for Figure 4b, we disable the *encoder training* in one or more stages, but always maintain the updates to actor and critic networks (e.g. S12 means encoder is frozen in stage 3).

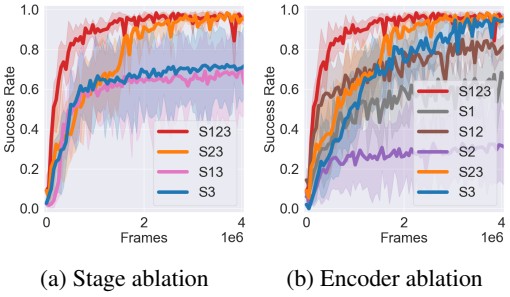

| (a) Stage ablation | (b) Encoder ablation |

Figure 4: (a) Effect of each training stage. The numbers after "S" indicates the stages that are enabled. (b) Effect of enabling encoder training in each stage.

Here we summarize a number of insights we can obtain from these results. In Figure 4a: Stage 1 pretraining improves sample efficiency (S123 > S23); **stage 2 is critical and its contribution cannot be replaced by the other stages** (S123> S13, S23 > S13). Note that since Adroit is challenging with limited offline data, S3 is always enabled to obtain non-trivial performance.

In Figure 4b: when the encoder is only trained in one stage, stage 3 encoder training gives a much stronger performance than the other two, showing the importance of task-specific representations (S3 > S1 and S2). Note that interestingly, **task-specific features are more useful, however, when RL**

**data is scarce, non-RL pretraining provides better results despite the large domain gap** (S3> S1 > S2). Training the encoder in more stages always leads to improved performance (S123 > S12 > S1).

An important finding is that our results indicate **offline RL is more beneficial than just contrastive representation learning in stage 2**, note that S3 in Figure 4b (perform offline RL update but freeze encoder in stage 2, encoder trained in stage 3) has much stronger performance than S3 in Figure 4a (no stage 2 training at all). This is consistent with results in Figure 2, the performance gap between VRL3 and FERM+S1 is caused by VRL3 doing offline RL updates in stage 2, and thus learning a full agent, instead of just the encoder features.

With the above insights, Table 1 summarizes fundamental differences between VRL3 and some other popular visual RL methods.

Table 1: Comparison of core design decisions. VRL3 fully utilizes non-RL, offline RL and online RL datasets. DA refers to data augmentation, Con for contrastive learning, Offline for offline RL. Note that offline RL data can be used to learn representation (encoder), or the task (actor, critic), or both.

| Characteristics | DrQ/RAD | CURL/ATC | FERM | RRL | VRL3 |
|---|---|---|---|---|---|
| Stage 1 | None | None | None | Pretrain | Pretrain |
| Stage 2 | None | None | Con | BC | Offline |
| Stage 3 | DA | Con | DA | No DA | DA |
| Leverage large visual datasets | ✗ | ✗ | ✗ | ✓ | ✓ |
| Task-specific representations | ✓ | ✓ | ✓ | ✗ | ✓ |
| Offline RL data for representations | ✗ | ✗ | ✓ | ✗ | ✓ |
| Offline RL data for task learning | ✗ | ✗ | ✗ | ✓ | ✓ |
| High data reuse rate (off-policy) | ✓ | ✓ | ✓ | ✗ | ✓ |
| Prevent Q network divergence | ✗ | ✗ | ✗ | ✗ | ✓ |

## 5.4 Additional Studies and Analysis

We also perform an extensive study on hyperparameter robustness and sensitivity on each of the 4 Adroit tasks (with varying difficulty). Due to limited space, we only show a small portion of our study in Figure 5 and put the rest of the results in Appendix A. This provides additional insights and helps us understand what hyperparameters should be tuned with high priority when applied to a new task, for example, results show that S1 pretraining is useful for harder tasks but not for the simplest Door task; data augmentation is always critical; the encoder learning rate is important and should be tuned in difficult tasks; even a small number of S2 offline updates can greatly help learning. In total, we studied 16 different components and hyperparameters, and identified 9 of them as important to performance, out of these 9, we further show that 6 of them are robust, and 3 of them are sensitive (learning rate, action noise std, encoder learning rate scale). Note that the encoder learning rate scale is **the only important and sensitive hyperparameter introduced by VRL3**, while the other 2 are from the backbone algorithm. These important studies show that our framework is quite robust, we believe these results and insights will provide a better understanding on data-driven DRL methods, and we hope they will make it easier for other researchers to deploy this framework to other tasks.

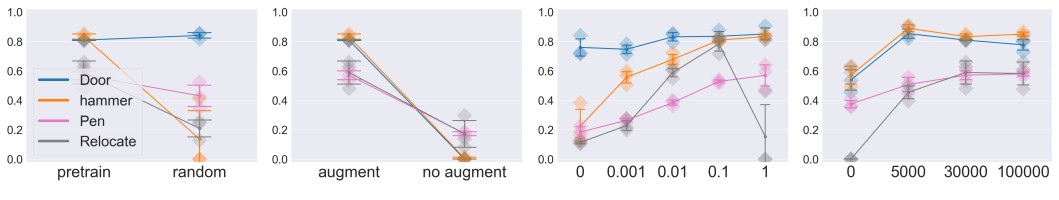

(a) S1 pretrain (R)  (b) S2,3 augmentation (R)  (c) S2,3 encoder lr scale (S)  (d) S2 num of update (R)

Figure 5: Part of the hyperparameter sensitivity study. For each figure, X-axis shows different hyperparameter values, Y-axis shows the average success rate in the first 1M training frames, over three seeds. Each dot is the average success rate of one seed, the error bar shows one standard deviation. (S) means critical and relatively sensitive. (R) means critical but is robust or easy to tune.

# 6   Discussion, Limitations and Future Work

In this paper, we propose a 3-stage data-driven framework for solving challenging visual control tasks. We perform extensive experimentation to ablate and analyze our framework to identify the most important technical components. We arrive at a method that is simple, intuitive, and much more sample efficient and computationally efficient than the previous SOTA on Adroit. Although this is a significant result, we do have some limitations: for example, Adroit has more realistic visuals compared to DMC, but it is still a simulated benchmark. Experiments on robotics hardware are required to understand how VRL3 should be deployed in the real world. Other limitations include: it remains unclear whether stage 1 training is still beneficial if the domain gap between non-RL data and RL data is too great (RL task visuals different from the real world). We also did not experiment extensively on model-based method. These can be exciting future research topics. Despite these limitations, VRL3 brings a number of novel and interesting insights, in addition to the strong empirical results. We believe VRL3 is an important step toward practical data-driven DRL for real world visual control tasks.

## Acknowledgments and Disclosure of Funding

The authors would like to thank Kan Ren and Tairan He for a number of interesting discussions on the effect of representation learning and pretraining. Thanks to Dongqi Han for helpful discussion and insights on potential encoder design and on the pretanh issue. Thanks to Yue Wang for an early discussion on the structure of the paper. Thanks to Han Zheng for insights on offline RL issues. Thanks to Kaitao Song, Xinyang Jiang, Yansen Wang, Caihua Shan, Lili Qiu for helpful feedback and discussions related to supervised learning and pretraining. Thanks to Zhengyu Yang, Kerong Wang, Ruoxi Shi, Bo Li, Ziyue Li, Yezhen Wang for discussion and feedback in group meetings. Thanks to Kexin Qiu for providing critical feedback on the naming of the framework. Thanks to Denis Yarats for the super clean DrQv2 codebase and answering some related questions. Thanks to Rutav Shah and Vikash Kumar for helping with the RRL codebase and Adroit on GitHub. Thanks to Suraj Nair for answering a question related to R3M computation speed. The authors also want to thank our reviewers and meta-reviewers for providing all the valuable feedback and suggestions.

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
