# A  Additional Results and Analysis

## A.1  Hyperparameter Robustness, Sensitivity and Understanding

In this section we discuss implementation details and present an extensive ablation study on hyperparameters.

Figure 6 shows how each hyperparameter affect training performance. For each figure, the results show how learning is affected when we modify one hyperparameter from our set of default hyperparameters while keeping all the rest fixed as the default values. Following our 3-stage framework, we discuss hyperparameters that affect one or more stages of our training process, and we also give our default value for each hyperparameter. The default hyperparameters are found in our early experiments with random search.

The goal here is to achieve a thorough understanding of what components are the most important for VRL3 to work, and when VRL3 is applied to a new task, which ones of the hyperparameters are the most critical and should be finetuned first. We believe these insights can greatly help researchers and practitioners in applying our framework to new visual tasks, or build research on top of our work. Note that in our study, we actually find some of the hyperparameters to be not important since they do not affect the training performance. However, they are still discussed here for completeness.

### A.1.1  Stage 1

**Stage 1: whether we use stage 1 pretraining**   (default: True). If we perform stage 1 pretrainig, then we start with an encoder that is pretrained on the ImageNet dataset, otherwise we start with a random encoder. This will only affect the initialization of encoder for stage 2 and 3. Figure 6a shows that the door task actually does not rely on stage 1 features, but the more challenging relocate task benefit hugely from stage 1 pretraining. This observation is consistent with previous results that show more difficult tasks benefit more from pretrained features, and easier tasks can be learned without them [73]. This result shows that stage 1 pretraining is critical especially if we want to tackle challenging real-world control tasks.

### A.1.2  Hyperparameters that affect both stage 2 and 3

During our training in stage 2 and stage 3, we can also add an additional BC loss, which can help the agent learn more human-like behavior [66]. We do not include it in VRL3 since it does not have a significant impact on performance, but will discuss it in this ablation study.

**Stage 2 and 3: whether we use data augmentation**   (default: True). Figure 6b shows that data augmentation is critical for both tasks. This result is consistent with previous literature that found data augmentation such as random shift can greatly improve performance in visual-based tasks [93, 94]. Although previous works mainly apply data augmentation to tasks with visuals that are different from the real-world such as DMControl[80], we show that it can also improves performance in tasks with more realistic visuals.

**Stage 2 and 3: demonstration replay batch size**   (default: 64). In stage 2, before the offline RL updates, we perform a number of BC updates where we sample from a demonstration replay buffer. In stage 3, we also do BC with a decaying loss. Figure 6g shows that the batch size for computing BC loss is not important and it is also not needed to achieve the performance VRL3. By default, we use a small batch size of 64 to reduce computation time.

**Stage 2 and 3: encoder learning rate scale**   (default: 0.01 for relocate, and 1 for the other tasks). Figure 6c shows that when encoder is fixed (learning rate scale is 0), then the performance becomes worse, especially for relocate. Note that when encoder learning rate is too high, learning can also become difficult. For relocate, we reduce its encoder learning rate to be 100 times slower than the learning rate for other parts of the agent, similar to what is done in [69]. This result, together with Figure 6a shows that when task-agnostic representation is finetuned towards task-specific representation learned directly from RL objective with data augmentation gives the best performance.

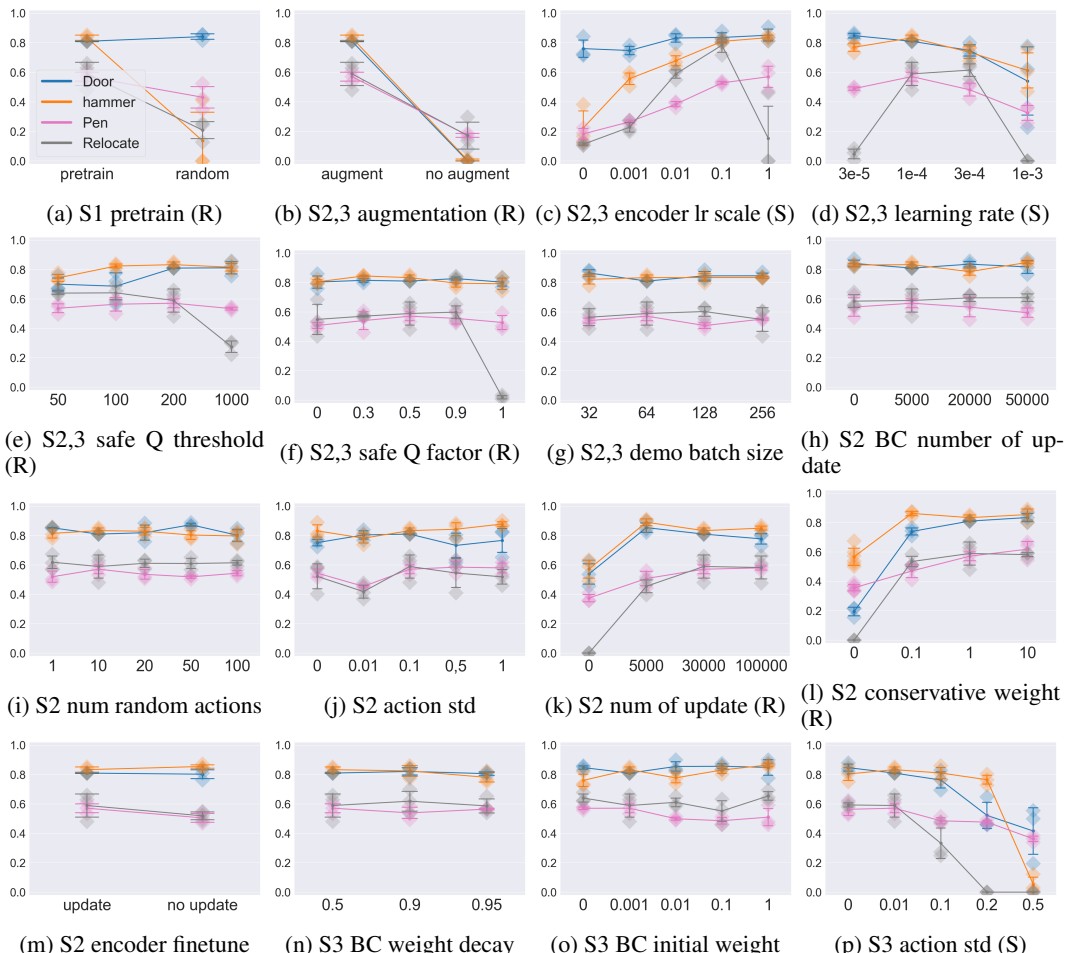

Figure 6: Hyperparameter sensitivity study for all three stages, blue shows the door task, orange shows the more difficult relocate task. For each figure, caption shows the type of hyperparameter, X-axis shows different hyperparameter values, Y-axis shows the average success rate in the first 1M frames of training, averaged over three seeds. Each dot is the average success rate of one seed, error bar shows one standard deviation. We use (S) to denote critical hyperparameter that is relatively sensitive and should be tuned with high priority when applied to a new task. (R) denotes critical hyperparameter that is robust or easy to tune. Hyperparameters that are mentioned in the main paper are discussed in the appendix.

**Stage 2 and 3: learning rate** (default: $1e-4$). Figure 6d shows that (unsurprisingly), when learning rate becomes too small or too large, performance will drop.

**Stage 2 and 3, safe Q target value threshold** (default: 200): Figure 6e shows that having a reasonable Q value threshold greatly helps improve performance on relocate. As described in the main text, This threshold decides how large the Q value can become. Having this safe Q target value technique allows easy control excessive Q bias accumulation with minimal negative effect.

**Stage 2 and 3, safe Q factor** (default: 0.5): Figure 6f shows that having a safe Q factor that is $< 1$ (when it is 1, then we are not limiting how large the Q values can be) can help stabilize training and prevent performance issues caused by Q divergence.

### A.1.3 Stage 2 hyperparameters

In stage 2, most of the hyperparameters here are about the offline updates. As mentioned in the previous section, we can have optional BC updates before the offline RL updates. For the offline RL updates, since we mainly use the idea of CQL, there are a number of CQL relevant hyperparameters.

**Stage 2, number of BC update**   (default: 5000): Figure 6h shows that BC updates in stage 2 before taking offline RL updates in fact does not affect performance too much. In terms of performance, it is not necessary when offline RL is applied at stage 2. However, it can still help the agent learn more human-like behaviors, as discussed in [66].

**Stage 2, CQL number of random actions**   (default: 10): in the author's implementation of CQL, a number of random actions are generated from a uniform disribution and these actions are used in the log-sum-exp computation along with actions proposed by the current policy. Though this is not really discussed in the CQL paper [37], the number of random actions might have an effect on offline training. Figure 6i shows that in our case, the number of random actions for CQL updates in stage 2 does not have a significant effect. However, we recommend future researchers to test this hyperparameter when applying VRL3 to new tasks, since the effect of this small detail is not fully addressed by the CQL authors.

**Stage 2, CQL action std**   (default: 0.1): since our off-policy backbone is DrQv2, we no longer have the entropy term in SAC, but use a fixed action std during updates and during exploration. We also test the effect of this std in stage 2 offline RL updates. Figure 6j shows that in our case, the standard deviation for the action distribution during CQL updates do not have a significant effect on training. Note that however, as we will discuss in the next subsection, this action std has a significant effect in stage 3 online learning.

**Stage 2, number of conservative updates**   (default: 30000): Figure 6k shows that when we do not use offline RL updates (0 updates), then BC updates in stage 2 alone cannot guarantee good performance in stage 3. Some of this difficulty has been studied in [58].

**Stage 2, conservative term weight**   (default: 1): this weight decides how hard the Q values are being affected by the conservative loss term [37]. Figure 6l shows that when CQL weight is greater than 0, the performance is in general robust with different values.

**Stage 2, allow encoder finetune in stage 2**   (default: True): by default, in stage 2 we finetune encoder with Q loss. Does this stage 2 finetuning have a significant effect on performance? Figure 6m shows that this does not affect performance too much in our case, indicating that for the encoder, stage 1 pretraining plus stage 3 finetuning is enough to obtain good performance. This shows in stage 2 we can do more than just using stage 2 data to learn representations. We can get the best performance by fully exploiting stage 2 data to learn an offline RL agent as well as improved representations,

### A.1.4 Stage 3 hyperparameters

**Stage 3, BC weight decay**   (default: 0.5): this value decides how fast the BC loss is decayed, following [66]. Figure 6n shows that under our current design of VRL3, the decaying speed of BC loss does not affect performance too much.

**Stage 3, BC initial weight**   (default: 0.001): The inital weight of the BC loss. Figure 6o shows that initial BC weight in stage 3 does not have a significant effect on performance. Again, we keep the BC loss here since it can also help the robotic hand learn human-like behaviors.

**Stage 3, action distribution standard deviation for exploration**   (default: 0.01): Figure 6p shows that we need a small std value to obtain good performance. This is likely because the Adroit tasks have sparse reward, thus it is easier to have less action noise so that the agent focus on exploring state space that is close to the expert demonstrations. This is another important hyperparameter that will require tuning on a new task. Note that in some other benchmarks such as DMControl, the exploration action noise is typically much larger.

## A.2 Hyperparameter Table

Since our backbone algorithm is DrQv2, we keep most of the original hyperparameters (see hyperparameter table of DrQv2 in appendix B of [93]). Table 2 shows a summary of the important hyperparameters used in VRL3. As discussed before, BC related hyperparameters show no effect on performance, and are removed from the table. To reshape rewards so that the maximum per-step reward is 1, the reward is divded by 100 for Hammer, 20 for Door, 50 for Pen, 30 for Relocate. So if we get a one-step reward of 100 in Hammer, it will be rescaled to 1.

For stage 1 our training is the same as in a typical ResNet training setting, we use an encoder with 5 convolutional layers (batch norm after each conv layer), with 32 channels in the first conv layer. For all 3 stages, we always use input size of 84x84. For stage 2, we use 25K offline RL data for both Adroit (expert demonstrations provided by the authors of the benchmark) and DMC (data collected by trained DrQv2 agents). When we move to stage 3, we are different from the default hyperparameters of DrQv2 in: a) we use a reduced encoder learning rate scale for relocate (DrQv2 does not discuss this option so it uses a value of 1). b) We do not collect random data at the beginning of stage 3, as we already have some offline data in stage 2. c) we use a fixed action std of 0.01 for exploration, while DrQv2 uses an exploration noise schedule.

For DMC tasks, we are largely the same as discussed in the DrQv2 paper [93], however, we use a fixed std of 0.1. We also found that a pretanh penalty can sometimes help avoid exploration issue, so we have a pretanh penalty weight (discussed in Section A.5) of 0.001 and a penalty threshold of 5.

Table 2: VRL3 default hyperparameters for Adroit

| Parameter | Value |
|---|---|
| *Encoder* | |
|     image input size | 84 x 84 |
|     n of channel in first conv layer | 32 |
|     number of convolutional layers | 5 |
|     batch norm | after each conv layer |
| *Stage 2* | |
|     offline RL data | 25,000 |
|     number of conservative updates | 30,000 |
|     CQL number of random actions | 10 |
|     CQL action std | 0.1 |
|     CQL weight | 1 |
| *Stage 2, 3* | |
|     optimizer | Adam [33] |
|     discount ($\gamma$) | 0.99 |
|     encoder lr scale | 0.01 for relocate, 1 for others |
|     learning rate | 1e-4 |
|     safe Q target threshold | 200 |
|     safe Q factor | 0.5 |
|     target smoothing coefficient ($\rho$) | 0.01 |
|     replay buffer size | $10^6$ |
|     mini-batch size | 256 |
|     nonlinearity | ReLU |
|     action repeat | 2 |
|     n-step returns | 3 |
|     agent update frequency | 2 |
|     encoder feature dimension | 50 |
|     Q and policy hidden layers | 2 |
|     Q and policy hidden dimension | 1024 |
| *Stage 3* | |
|     random starting data | 0 |
|     action std for exploration | 0.01 |

## A.3 Additional Figures

Here we present per-task figures for results in the main paper, for each set of figures, additional discussions are provided in the caption.

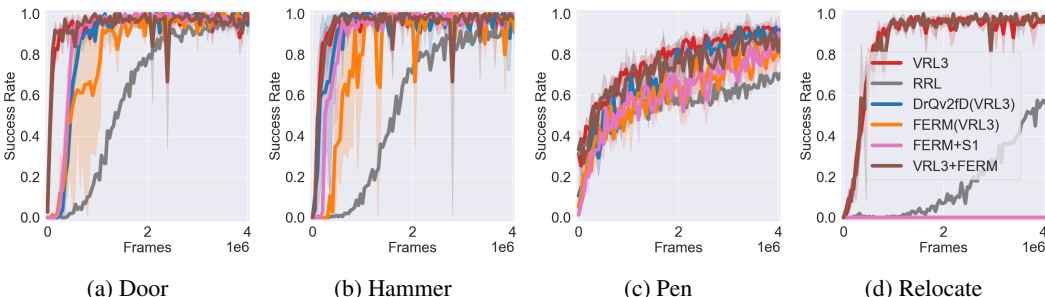

(a) Door  (b) Hammer  (c) Pen  (d) Relocate

Figure 7: Per-task success rate comparison of VRL3 and other methods. Note that VRL3 is most effective on the hardest Relocate tasks (Prior to VRL3, RRL is the only method that achieves non-trivial performance on Relocate, but takes more than 12M frames to reach 90% success rate, while VRL3 solves it in 1M frames), but also significantly outperform other variants on Door and Hammer. VRL3 outperforms RRL on Pen, but is similar to the other competitors in sample efficiency.

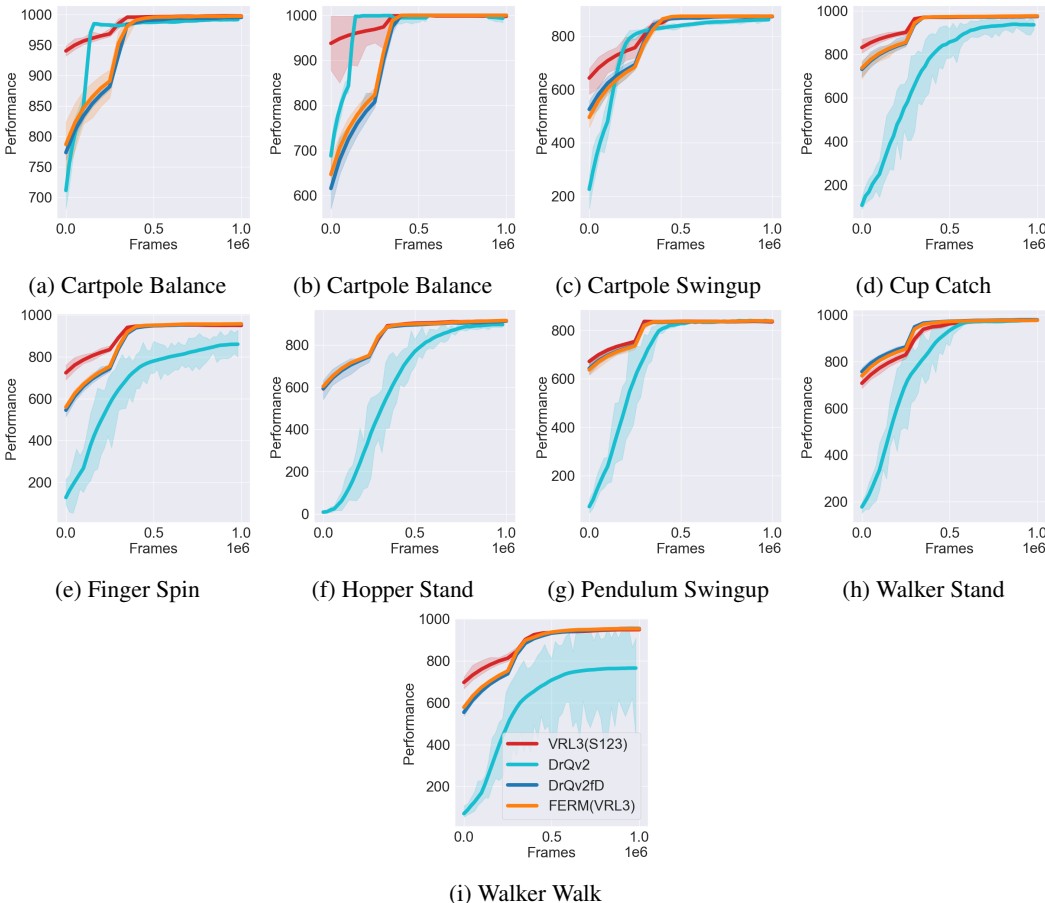

(a) Cartpole Balance      (b) Cartpole Balance      (c) Cartpole Swingup      (d) Cup Catch

(e) Finger Spin      (f) Hopper Stand      (g) Pendulum Swingup      (h) Walker Stand

(i) Walker Walk

Figure 8: Per-task performance comparison of VRL3 and other methods on all 9 Easy DMC tasks. The methods that utilize stage 2 data (VRL3, FERM, DrQv2fD) tend to learn faster. Overall, VRL3 has a small advantage over DrQv2 and FERM, but the performance gap is smaller compared to the results on Medium/Hard DMC tasks and the more challenging Adroit tasks. For VRL3, FERM and DrQv2fD, We use the same hyperparameter setting for all tasks. The DrQv2 results are from the authors.

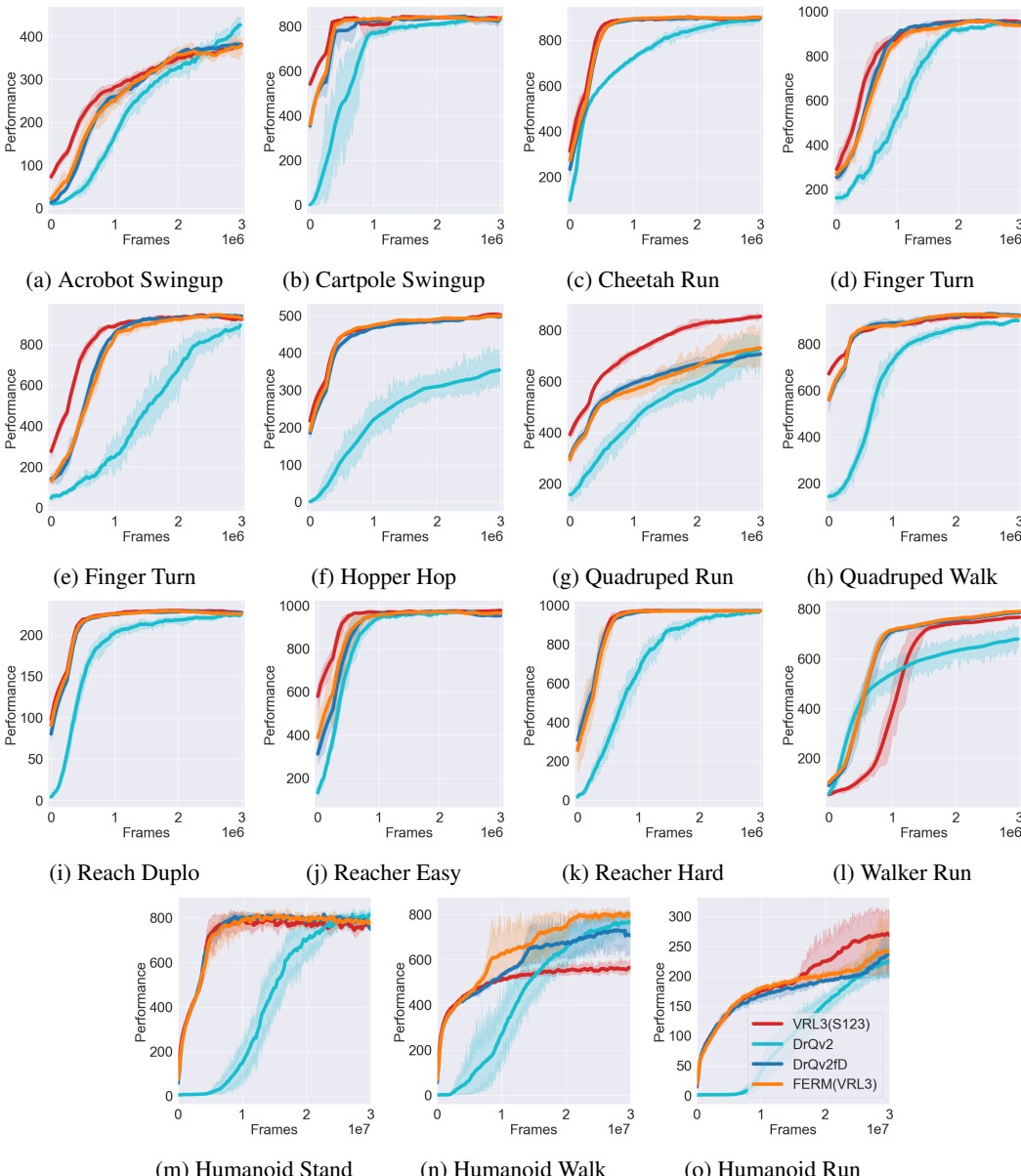

Figure 9: Per-task performance comparison of VRL3 and other methods on all 15 Medium and Hard DMC tasks. The methods that utilize stage 2 data (VRL3, FERM, DrQv2fD) tend to learn faster. Overall, VRL3 has an advantage over DrQv2 and FERM in some environments, but overall, their performances are similar. This again shows that the advantage of multi-stage training procedures such as VRL3 is most prominent in the most challenging tasks with sparse reward and realistic visual input. For VRL3, FERM and DrQv2fD, We use the same hyperparameter setting for all tasks (they are the same for all 24 tasks). The DrQv2 results are from the authors.

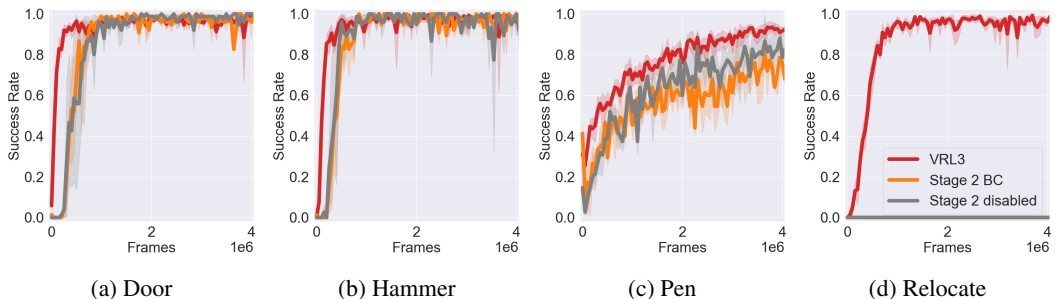

Figure 10: Per-task success rate comparison for VRL3, VRL3 with BC in stage 2 instead of conservative updates (Stage 2 BC), and VRL3 with stage 2 entirely disabled (Stage 2 disabled). Results show that using conservative RL updates in stage 2 is important, especially in Relocate. In the other three tasks, we also observe a significant performance gap in the early stage of training: VRL3 is able to quickly achieve a high performance with small amounts of online data, while the other 2 variants take much longer to reach the same level of performance.

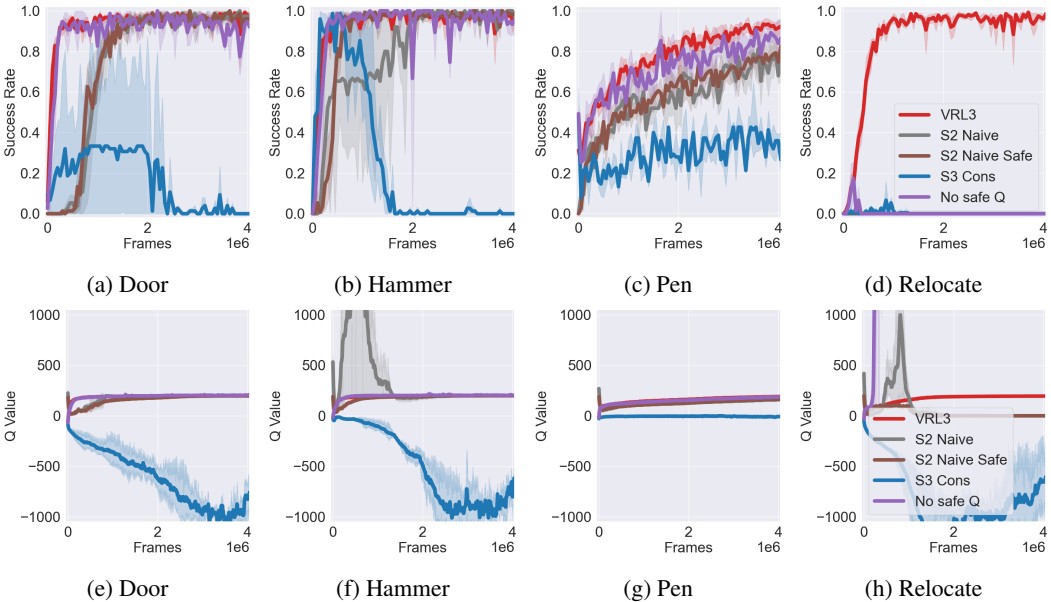

Figure 11: Per-task success rate (first row) and Q value (second row) comparison for VRL3 with different stage 2 to stage 3 transition methods. VRL3 by default uses conservative updates in stage 2, standard Q updates (without the conservative term) in stage 3, and uses safe Q in both stages 2 and 3. VRL3 has the best performance and the most stable Q values throughout stage 3 training. Each of the variants shown here is different from VRL3 in just one aspect. We can see that using naive RL updates in stage 2, and disable Safe Q lead to significantly reduced performance in all 4 tasks, and from the second row we see a significant overestimation issue. Note that using naive RL in stage 2 but enabling Safe Q leads to better performance in Hammer, and a now stable Q value in all four tasks. This shows that Safe Q is effective in stabilizing Q value estimates, but it cannot replace the role of proper stage 2 training. S3 Cons uses conservative learning in stage 3 instead of standard Q learning, giving the worst performance, and a significant underestimation of Q values in all tasks. And if we use conservative updates in stage 2, but disable Safe Q (No safe Q), then we can learn well in the easier 3 tasks: Door, Hammer and Pen. But in Relocate we run into severe overestimation and the agent fails entirely. This shows that although Safe Q might not be needed in easier tasks, it is critical in more challenging tasks. Note that Safe Q is not the only technique that can achieve this result, and we use it because we are aiming for a minimalist design, and Safe Q is easy to understand, to implement, and has negligible computation overhead.

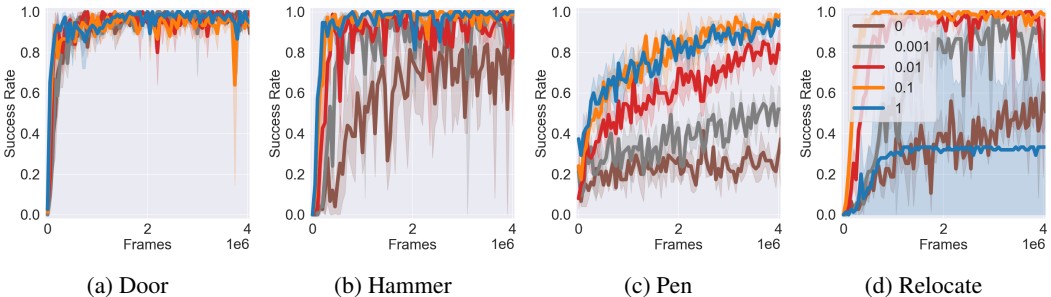

(a) Door        (b) Hammer        (c) Pen        (d) Relocate

Figure 12: Per-task success rate comparison for VRL3 with different encoder learning rate scales. Notice that in the easiest Door task, using an encoder learning rate scale of 0 (which means the encoder is pretrained in stage 1, and frozen afterwards) can still work. But in the more challenging tasks, tuning the encoder learning rate scale leads to much better results. In the most challenging Reloate task, note that the best encoder learning rate is 0.1 and 0.01: the encoder is being learned 10 or 100 times slower than rest of the agent (the policy and the Q networks).

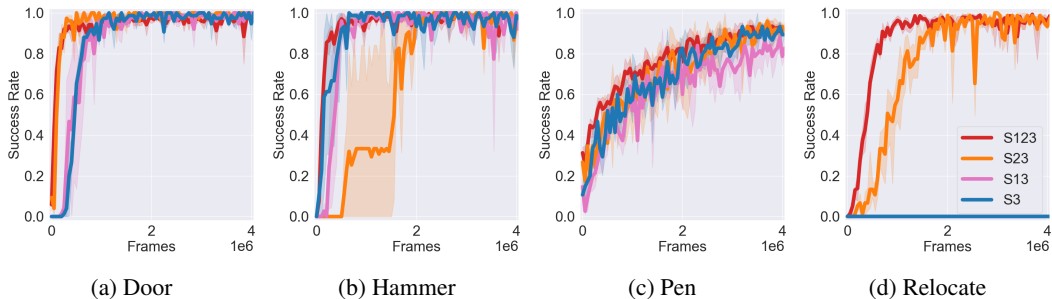

(a) Door        (b) Hammer        (c) Pen        (d) Relocate

Figure 13: Per-task success rate comparison for VRL3 with different training stages enabled. Stage 1 pretraining is important for the Hammer and Relocate tasks (S123 > S23), while having minimal effect on the easiest Door task. Stage 2 is consistently important in all tasks (S123 > S13). Note that interestingly, stage 1 plus stage 3 is not always stronger than stage 3 alone (S13 < S3 on Pen). This shows that stage 1 pretraining should be treated carefully when there is a domain gap. Note that when all three stages are combined, we consistently get the best performance on all environments.

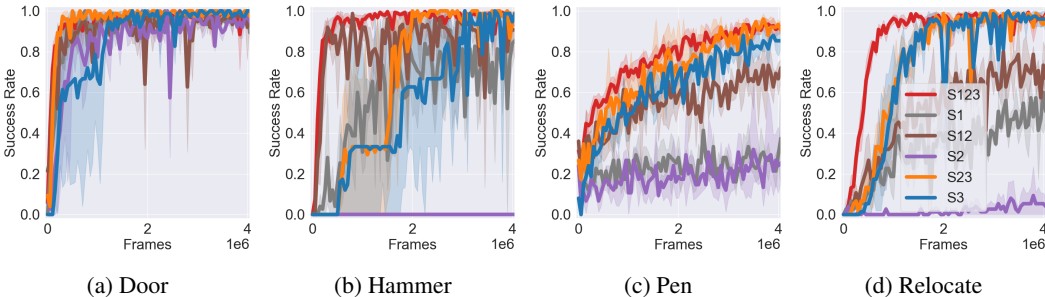

(a) Door        (b) Hammer        (c) Pen        (d) Relocate

Figure 14: Per-task success rate comparison for VRL3 with encoder training enabled in different stages. Note that since in stage 2, we only perform a small number of offline updates (30K updates) on a limited amount of RL data, stage 2 (S2) alone is not enough to obtain good representations for efficient learning (Although the Door task is easy and can be learned in all settings). Stage 1 (S1) alone is also not sufficient and gives bad performance, but combining stage 1 and 2 (S12) gives much stronger performance. Stage 3 alone also gives sub-optimal performance, especially on Hammer and Relocate. Compared to Figure 13, the setting in this figure isolates the effect of encoder feature learning from RL training.

## A.4 Finetuning batch norm layers

When we finetune the convolutional encoder, one issue that is rarely studied in the RL setting is how the batch norm layers should be treated. In our case, by default, we set the batch norm layers to be in evaluation mode and disable further gradient updates for more stability. However, we also provide a set of ablations to show that in Adroit, these options are not really affecting performance too much. Figure 15 shows the results. We can see that all four variants give similar performance.

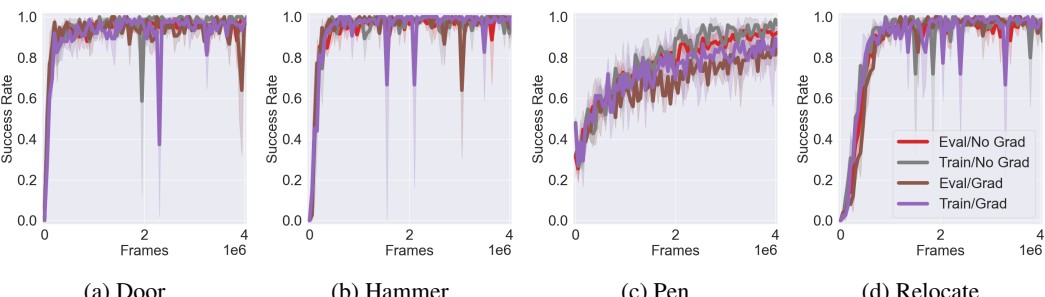

| (a) Door | (b) Hammer | (c) Pen | (d) Relocate |

Figure 15: Per-task success rate comparison for VRL3 with different batch norm training settings. Train and Eval indicate whether during finetuning in the stages 2 and 3, the batch norm layers are in train or evaluation mode (in evaluation mode, batch norm will use running statistics and not the standard per-batch statistics). No Grad means the batch norm layers will not be further updated with gradients from the RL objective.

## A.5  Pretanh Penalty

Another issue that we considered (but is not discussed in the main paper) is whether the tanh activation function in the policy network might saturate during training. Note that since Adroit and DMC are tasks with continuous and bounded action spaces, a tanh function is typically used to bound the output action value from the policy network. So the "pretanh value" refers to the action value before going through the tanh activation. Note that if the absolute value of the pretanh value is too large then no gradient can be propagated back. We found that applying a penalty on the average absolute pretanh value can effectively prevent tanh saturation in the policy network. Though in terms of performance, it does not have a significant effect in Adroit and thus is not necessary. In DMC we found it occasionally can prevent exploration issues, so we apply a penalty weight of 0.001 in all DMC tasks. To implement the penalty we essentially add an L2 regularization term on the pretanh value to the policy loss. To avoid affecting the agent learning when unnecessary, we set a threshold value of 5 so that the penalty only applies when absolute pretanh values are larger than this threshold. Details are provided in our source code.

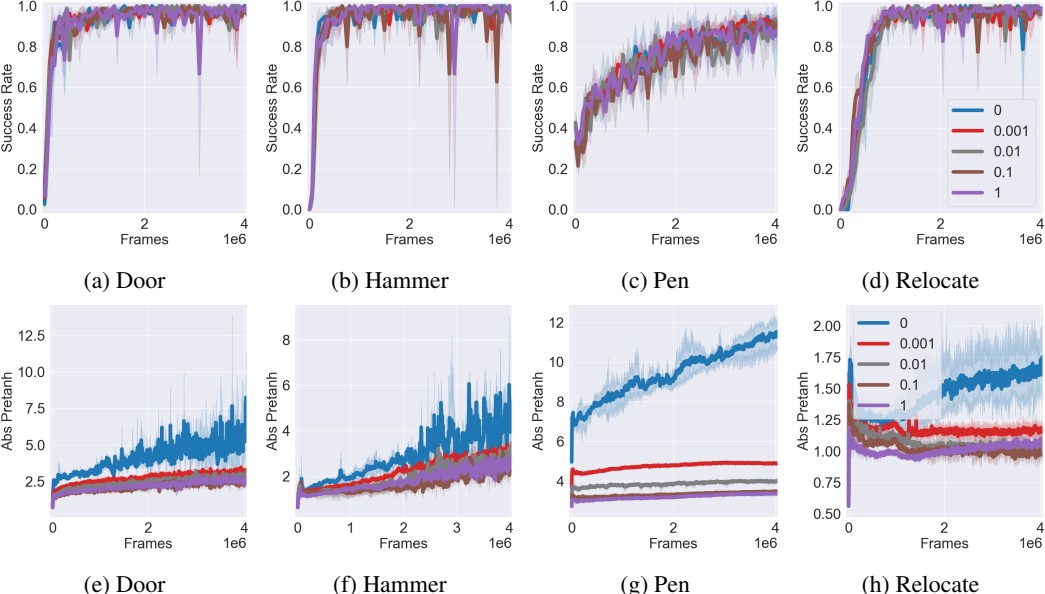

Figure 16: Per-task success rate comparison for VRL3 with different pretanh penalty weights, first row shows success rate, second row shows average absolute pretanh value. We can see that when we do not apply a pretanh penalty, the average absolute value of the pretanh values become fairly large, indicating tanh saturation. Although in the case of Adroit tasks, they seem to not affecting performance too much (This might be due to the particular implementation design in our backbone algorithm: the action noise is injected after the tanh activation. Although this design cannot prevent tanh saturation, it can in fact allow better exploration when tanh is saturated). Related issues have been studied in the past [26, 86], however, it might be worth it to revisit this important problem in future research.

## A.6    Comparing to RRL Long-term Performance

Figure 17 shows that in terms of the long-term performance, VRL3 and RRL (trained until 12M, data provided by RRL authors) achieves similar results, VRL3 is slightly stronger in Relocate, and in Pen, VRL3 reaches 90% success rate at 3.25M data and then continues to increase, RRL fails to reach 90% success rate even at 12M data. Based on these numbers, it is fair to say that VRL3 outperforms RRL in both early stage and late stage. But note the long-term performance difference is smaller. That is why in our main text, we focused on discussing the improvement in sample efficiency (measured as the number of data needed to reach 90% or more success rate), compared to RRL, we are on average 780% better. In appendix B.1 we provide details on how we compute these numbers and also provide comparison on how fast we are to reach 50% and 75% success rate. In all cases, VRL3 has a significant advantage.

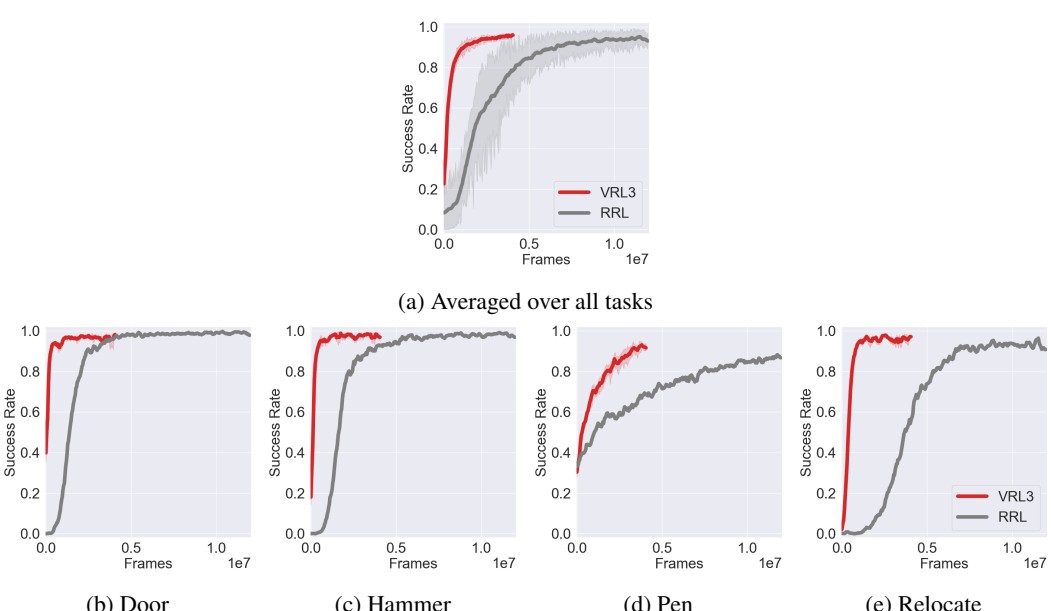

Figure 17: Success rate comparison for VRL3 at 4M and RRL at 12M. Results show that the long-term performance of RRL and VRL3 are similar. The advantage of VRL3 is mainly on sample efficiency.

## A.7 Using Deeper and Wider ResNet Encoders

For the majority of our results, we use a shallow encoder with 5 convolutional layers. This is consistent with many prior works in pixel-input DRL research [35, 93, 43, 55, 56, 84] and also makes computation faster. However, it is also beneficial to know whether our method can work with deeper and wider ResNet encoders. In Figure 18 we show that, on the hardest Relocate task, we can achieve similar performance with deeper and wider encoders. Here we study 6 different encoders: ResNet6 with 32 channels (default for VRL3), ResNet6 with 64 channels, ResNet10 with 31 channels, ResNet10 with 64 channels, ResNet18 with 64 channels, ResNet18 with 64 channels. For ResNet10 and ResNet18, we have skip connections as in typical ResNets. Results show that VRL3 can still learn effectively with these encoders of different capacity. However, note that ResNet18 achieves weaker performance, likely due to the fact that we are using the default hyperparameters. Since these encoders have much higher capacity, it might be possible to achieve better performance by finetuning the learning rate. Also note that when using ResNet6 with 64 channels, we observe an even better performance: it is able to reach 90% success rate on the hardest Relocate task with only 0.45M data, making it 24 times more sample efficient compared to the previous SOTA.

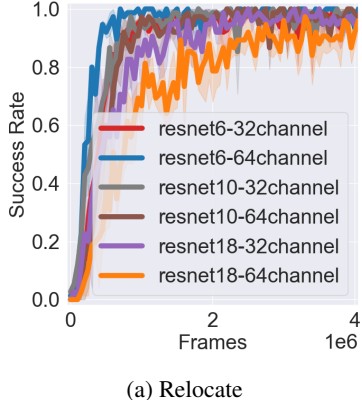

(a) Relocate

Figure 18: Success rate comparison for VRL3 with different ResNet encoders on the hardest Relocate task, all using the same default hyperparameters. When increasing the capacity of a ResNet6 encoder, we can obtain even stronger performance.

In Figure 19, further experiments show that finetuning the encoder learning scale can allow effective learning with the deeper ResNet10 and ResNet18 encoders (both with 32 channels) in both the easier Door task and the harder Relocate task. Note that when using a deeper encoder, the agent seems to become more sensitive to the encoder learning rate scale. In our main results, we did not use deeper networks since the shallow 5-layer network already provides strong performance, and is more consistent with the network size used in prior pixel-input DRL works.

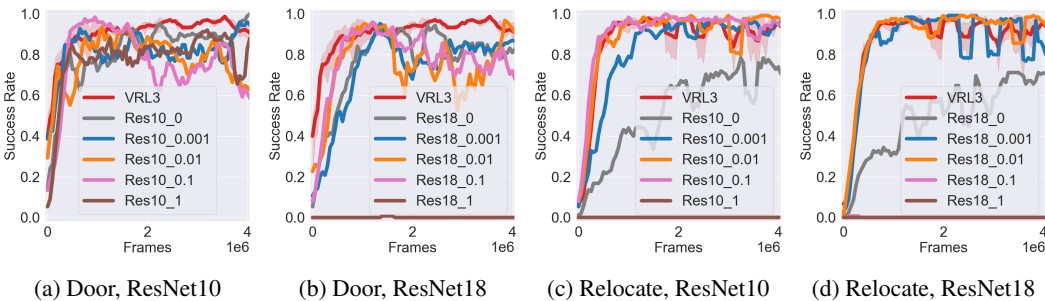

(a) Door, ResNet10    (b) Door, ResNet18    (c) Relocate, ResNet10    (d) Relocate, ResNet18

Figure 19: Success rate comparison of VRL3 with ResNet10 and ResNet18 encoders and with different encoder learning rate scale.

## A.8 Alternative BC Baselines

In Figure 20 and Figure 21 we present results for two other BC baselines: a) In stage 2, we perform BC updates, and we also train the value network to predict discounted cumulative reward of the offline trajectories (we call this "BC MC"). b) we train the value network with a slightly different method, we perform the standard Q-learning updates, but when computing the Q target, we replace the "next action from current policy" with the actual next actions in the offline trajectories, so as to circumvent the extrapolation error issue (we call this "BC Offline Act"). We then compare these 2 new variants with VRL3 (uses offline RL in stage 2) and BC naive (the original BC baseline that only learns the policy network and not the Q networks). Additionally, we also compare the performance of these 4 variants when stage 1 pretraining is enabled (Figure 20), or disabled (Figure 21).

The result is quite interesting: First note that VRL3 has the strongest overall performance, especially on the most difficult Relocate task. And stage 1 pretraining has a quite significant effect on performance, note that in Door and Hammer, the 3 BC baselines perform similar to VRL3, even though their stage 2 training is quite naive. And when stage 1 pretraining is disabled in Figure 21, even though the performance of VRL3 also drops, the performance gap is now much larger. This seems to show that the problems in offline-online RL transition can be mitigated by stage 1 pretraining.

For the 3 BC variants, note that none of the variants can consistently outperform others. For example, BC Offline Act is better than BC MC in Door and Hammer, but a bit weaker in Pen. Also note that surprisingly, when there is no stage 1 pretraining, sometimes BC Naive can work much better than other BC baselines. We did not expect this result because intuitively a somewhat decent policy-value match should outperform a policy with untrained value function.

A potential explanation is that if we train the value network in a naive way (BC Offline Act and BC MC) in stage 2, this might inject harmful inductive bias into the value network, making it hard for learning in the later stage. There have been a few recent papers that discuss this issue: for example, [60] show that early stage training, especially on a small dataset can impose a permanent effect on RL networks and reduce their long-term performance. It is possible that in some cases, an untrained Q network learns better in stage 3 compared to a naively trained Q network, because it is free from the inductive bias generated by the naive value learning process. In a related work, [53] show that networks gradually lose capacity during DRL training, and propose to maintain this capacity with a simple regularization. Further investigation in this direction might lead to interesting future work.

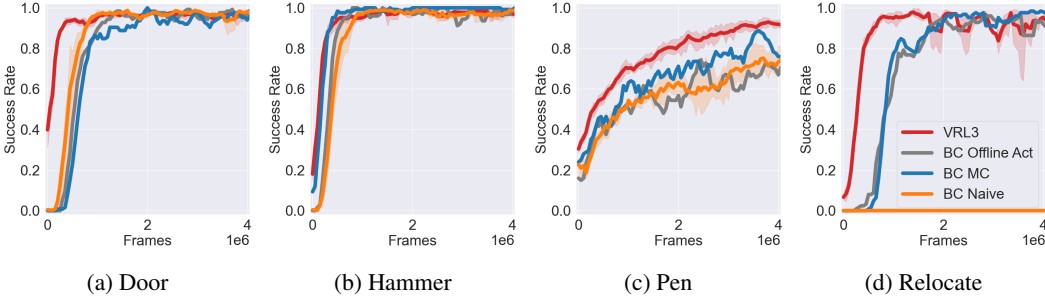

(a) Door      (b) Hammer      (c) Pen      (d) Relocate

Figure 20: Success rate comparison of VRL3 and three BC baselines. Stage 1 pretraining enabled.

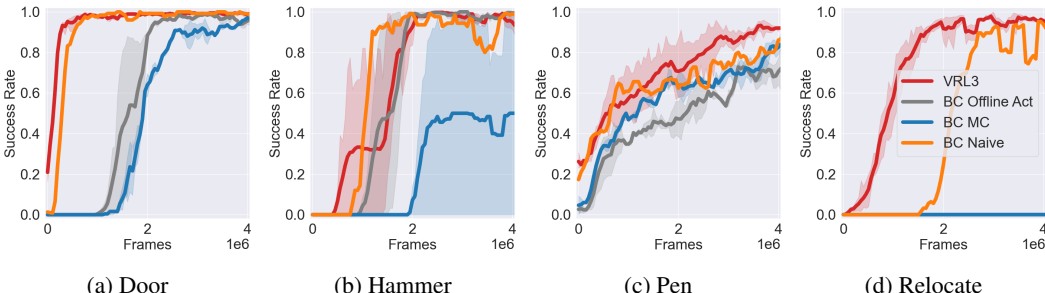

(a) Door      (b) Hammer      (c) Pen      (d) Relocate

Figure 21: Success rate comparison of VRL3 and three BC baselines. Stage 1 pretraining disabled.

## A.9 CCE Ablations

In Figure 22 we present ablation on CCE and compare to four variants: 1) We initialize the first encoder layer randomly (Rand First). 2) We treat each input image separately, and then sum their encoding (Encoding Sum) 3) Similar to 2), but we concatenate their encoding (Encoding Concat) [59, 62, 90], 4) similar to 2), but we also concatenate the difference of image encodings, according to [74]. Our newest results show that the overall difference in performance among these variants is actually quite small, and we do not have a particular variant that can consistently outperform others. However, note that for the 3 variants where we sum or concatenate image encodings, there will be a significant computation overhead. On a V100 GPU, for the Relocate task, 1000 updates for CCE/Rand First take about 47 seconds, while for Encoding Sum, Encoding Concat and Encoding Diff Concat, they take about 200 seconds, about 4 times slower than using CCE (other parts of VRL3 are kept the same during this comparison). Note that the Rand First also obtains surprisingly good performance. Intuitively, this should not give a good performance since the first layer of the pretrained encoder is entirely reset. It might be that useful features in the encoder are being recovered during stage 2 training, or perhaps the Adroit benchmark does not require very fine-grained features in the encoder, further investigation might be required here.

Given these results, it seems CCE is still the best fit for our framework, because it is intuitive, easy to use (can be done in one line of code), and has no computation overhead.

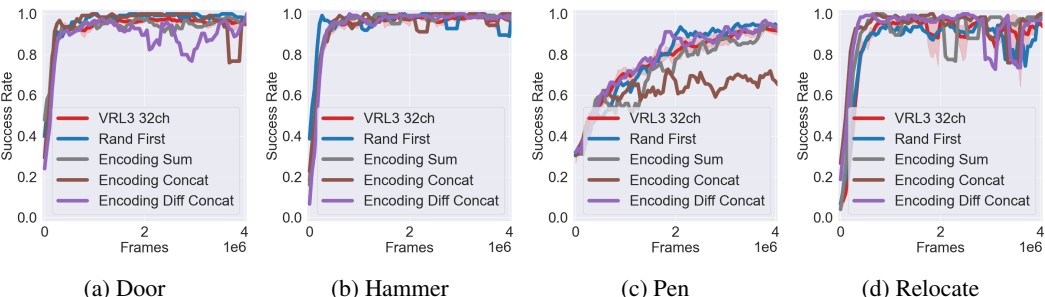

(a) Door      (b) Hammer      (c) Pen      (d) Relocate

Figure 22: Success rate comparison of CCE and 4 other variants.

## A.10    Additional Stage 1 Pretraining Schemes

In Figure 23, we show that alternative ways to perform stage 1 pretraining can also lead to better performance compared to when no stage 1 pretraining is used. For all 3 contrastive pretraining variants, we employ a training scheme similar to BYOL [21], and we use a momentum encoder to generate positive pairs. For Contrast, we use random crop and resize, random gray scale, color jittering as augmentations; For Contrast-all, we additionally add horizontal flips; For Contrast-shift, we only use random crop and resize, this is a bit similar to the random shift augmentation used in DrQ/DrQv2. Each variant receives 60 epochs of contrastive pretraining on ImageNet. It is interesting that Contrast-shift seems to have the best performance, out of the 3 contrastive stage 1 variants. All 3 contrastive variants perform a bit weaker than the default scheme.

These results are not extensive and are only meant to show that other alternative methods also work. It does not indicate that classification pretraining is the best for DRL. In our case, it is possible that using larger datasets for contrastive pretraining, finetuning hyperparameters, or training for more epoch might lead to better results. And in fact, a few recent papers have shown that unsupervised pretraining can provide very useful features for control [59, 62].

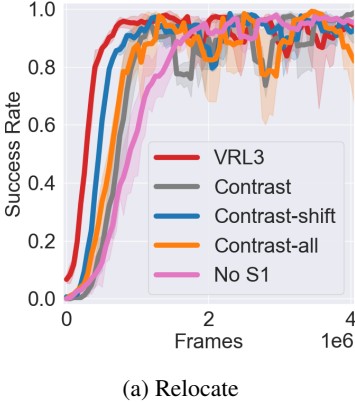

(a) Relocate

Figure 23: Success rate comparison for VRL3 (default ImageNet classification stage 1 pretraining) and VRL3 variants with alternative contrastive stage 1 pretraining.

# B  Efficiency Comparison

## B.1  Sample Efficiency Comparison

Table 3 shows a comparison of VRL3 and RRL in terms of sample efficiency. Sample efficiency is important because online data can be hard and expensive to obtain in DRL control tasks. Being able to reach a high sample efficiency means we can solve the task faster and with much less cost. Results show that VRL3 has much better sample efficiency in all tasks, giving an average of 780% better sample efficiency in reaching 90% success rate in all Adroit tasks, with an impressive 1220% better sample efficiency on the most challenging relocate task. If we consider the additional result with doubling the number of channels for Relocate (Appendix A.7), then VRL3 can learn Relocate in 0.45M, making it 2440% times more sample efficient than RRL.

Table 3: Sample efficiency comparison of VRL3 and RRL. The numbers show the number of data (frames) collected when the specified success rate is achieved. (to be precise, the frame number indicates the agent has an average success rate that is equal or greater than the specified success rate for the past 20K frames of training.) The last column show how many times VRL3 is more sample efficient than RRL in reaching that success rate. VRL3 is efficient in learning especially when aiming for a high success rate. Averaged over all Adroit tasks, we are 780% more sample efficient than RRL, the previous SOTA method, to reach 90% success rate. To ensure our performance is not the result of "lucky random seeds"[27, 28, 15], the performance for VRL3 reported in this table is averaged over 10 seeds. To ensure consistency with prior work, results for RRL are directly from the training log files provided by the authors. Note in the pen task, RRL's success rate at 12M is slightly less than 90%, but close.

| Score | RRL | VRL3 | VRL3 faster |
|---|---|---|---|
| Door at 50% | 1360K | 200K | 6.8 |
| Door at 75% | 1880K | 300K | 6.3 |
| Door at 90% | 2560K | 400K | 6.4 |
| Hammer at 50% | 2080K | 250K | 8.3 |
| Hammer at 75% | 3120K | 350K | 8.9 |
| Hammer at 90% | 4440K | 500K | 8.9 |
| Pen at 50% | 1120K | 400K | 2.8 |
| Pen at 75% | 6200K | 1600K | 3.9 |
| Pen at 90% | 12000K | 3250K | 3.7 |
| Relocate at 50% | 4360K | 500K | 8.7 |
| Relocate at 75% | 6800K | 650K | 10.5 |
| Relocate at 90% | 11000K | 900K (450K) | 12.2 (24.4) |

## B.2 Parameter Efficiency Comparison

Table 4 gives a comparison of number of parameters for VRL3 and RRL. Note that in addition to the encoder, other networks in the agent can also have a lot of parameters, so we have to approach this comparison carefully. Note that VRL3 has a much smaller encoder and relatively larger MLP critic and actor networks. If we count all the parameters in the agent, then VRL3 has about 37% of the pararmeters as used in RRL with a ResNet34. A benefit of having less parameters is that the agent will use less memory, making it easier to deploy on portable devices.

Note that being parameter-efficient is also not exactly the same as being computationally efficient. In the next section, we discuss computation efficiency.

Table 4: Parameter efficiency comparison of RRL and VRL3. We compare the number of parameters in the encoder part, as well as parameters for all networks in the agent. Note that VRL3 has a much smaller encoder, but relatively larger critic and policy (actor) networks, VRL3 also has a pair of critic networks instead of just one. For encoder, VRL3 is more than 50 times more parameter efficient. Overall, VRL3 is about 3 times more parameter efficient.

| Number of parameters (M) | RRL | VRL3 | VRL3/RRL ratio |
| --- | --- | --- | --- |
| Encoder | 21.28 | 0.40 | 0.0187 |
| Policy network | 2.0 | 3.0 | 1.5 |
| Critic networks | 2.0 | 6.0 | 3 |
| All | 25.28 | 9.4 | 0.37 |

## B.3 Computation Efficiency Comparison

We now compare the computation efficiency of RRL and VRL3 . We do not take into account the time used for agent evaluation (in which case RRL is much slower due to the large ResNet34 encoder) for RRL we use the authors' codebase. On a single NVIDIA V100 GPU, for door, RRL runs at 83 FPS while VRL3 runs at 68 FPS; for relocate, RRL runs at 53 FPS while VRL3 runs at 47 FPS.

Note that RRL is faster in per-frame computation mainly because its encoder is frozen, so images only need to be processed by the encoder once when they are collected. For VRL3, although the encoder is 50 times smaller, the encoder will be updated in all stages, and the off-policy method we are based on also use a larger batch size and larger actor and critic networks.

Note that although VRL3 will take a bit longer time to train for the same number of frames, it is significantly faster when we consider the fact that it can learn a task with a small amount of online data. In terms of the computation time used to reach 90% success rate, VRL3 is much faster on all tasks, and more than 10 times faster on the most challenging relocate task. Note that here we are not considering acceleration from parallel or multi-core computing.

Table 5: Computation efficiency comparison of RRL and VRL3.

| Computation Time (Hours) | RRL | VRL3 | VRL3/RRL ratio |
| --- | --- | --- | --- |
| Door, Pen, Hammer (4M) | 13.39 | 16.34 | 1.22 |
| Relocate (4M) | 20.96 | 23.64 | 1.13 |
| Door (90% success) | 8.57 | 1.63 | 0.19 |
| Hammer (90% success) | 14.86 | 2.04 | 0.14 |
| Pen (90% success) | 40.16 | 13.28 | 0.33 |
| Relocate (90% success) | 57.65 | 5.32 | 0.09 |

## B.4 Computing Infrastructure

Our experiments are conducted on NVIDIA Tesla P40, P100 and V100 GPUs. When calculating the computation efficiency table, we use a single V100 GPU.

# C   Other Technical Details

## C.1   DMC, Adroit and MuJoCo Usage

To use the MuJoCo[82] physics simulator, please refer to `https://mujoco.org/`, as of the writing of this work, MuJoCo is now free and a license is not required. For DMC[80] please refer to `https://www.deepmind.com/publications/deepmind-control-suite`. For Adroit [66] please refer to `https://github.com/facebookresearch/RRL` and `https://sites.google.com/view/deeprl-dexterous-manipulation`.

## C.2   DrQv2 Codebase

Our backbone algorithm is DrQv2[93], the code can be found at `https://github.com/facebookresearch/drqv2`. The code is under MIT License.

## C.3   Stage 1 Pretraining

Our code for stage 1 pretraining largely follows the example code given here: `https://github.com/pytorch/examples/blob/main/imagenet/README.md`.

## C.4   ImageNet Dataset

The training ImageNet data can be found here: `https://www.kaggle.com/competitions/imagenet-object-localization-challenge`, to organize the validation data, we use the script here: `https://raw.githubusercontent.com/soumith/imagenetloader.torch/master/valprep.sh`

## C.5   Algorithm Psuedocode

For better clarity, in the main paper we provide a high-level description of our framework. Here we also provide the psuedocode.

---

**Algorithm 1** Visual DRL in 3 stages

---

1: Initialize parameterized encoder $f_\xi$, policy $\pi_\phi$, Q functions $Q_{\theta_1}, Q_{\theta_2}$, Q target functions $Q_{\bar{\theta}_1}, Q_{\bar{\theta}_2}$, empty buffer $\mathcal{D}$. Set target parameters $\bar{\theta}_i \leftarrow \theta_i$, for $i = 1, 2$. $N_{S2}$ is the number of offline RL updates in stage 2; $N_{S3}$ number of online RL updates for stage 3.
2: Stage 1:
3: Load pretrained encoder parameters into $\xi$ and setup encoder for RL training.
4: Stage 2:
5: Load offline RL data into $\mathcal{D}$
6: **for** $N_{S2}$ updates **do**
7:     Sample a mini-batch $B$ from $\mathcal{D}$
8:     Update $f_\xi$ and $Q_{\theta_1}, Q_{\theta_2}$ with $\mathcal{L}_Q + \mathcal{L}_{QC}$, update $\pi_\phi$ with $\mathcal{L}_\pi$. Update $Q_{\bar{\theta}_1}, Q_{\bar{\theta}_2}$ with polyak averaging.
9: Stage 3:
10: **for** $N_{S3}$ updates **do**
11:     Sample an action from policy, take action and store the new data in $\mathcal{D}$
12:     Sample a mini-batch $B$ from $\mathcal{D}$
13:     Update $f_\xi$ and $Q_{\theta_1}, Q_{\theta_2}$ with $\mathcal{L}_Q$, update $\pi_\phi$ with $\mathcal{L}_\pi$. Update $Q_{\bar{\theta}_1}, Q_{\bar{\theta}_2}$ with polyak averaging.

---

# D  Further Discussions

## D.1  Additional Related Work on Bias Reduction

Overestimation and underestimation bias issues have been studied in a large number of previous works, and has long been identified as one of the major obstacles in off-policy learning [81]. A number of important works have achieved significant performance gain over previous methods mainly with better bias control, For example, DDQN [84] achieves much stronger performance over DQN[55, 56] with double Q learning, TD3[19] greatly improve robustness of DDPG[51] with Clipped Double Q (CDQ). CDQ is also used in other popular methods such as SAC[23] and DrQ[35, 93]. Other methods such as weighted Bellman updates can reduce bias propagation[45], multi-step methods can also help[54]. Recently, it has been shown that the underestimation issues can also occur in DRL training and be problematic [41], finer-grained bias control can further improve performance [39], and ensemble-based bias reduction can exploit an increased number of network updates and greatly improve sample efficiency [7]. Note that the bias problem is a well-established issue that has been studied in many papers and this is not an exhaustive list.

Fundamentally, the Safe Q technique is another bias reduction technique that helps training by posing a soft constraint on the Q value estimates. The stage 2-3 transition scenario is a bit similar to the increased network updates setting as studied in [7], where the bias issue is exacerbated, and thus additional bias control is required. Note that VRL3 uses Safe Q on top of CDQ. Figure 11 essentially shows that CDQ alone (all variants in the figure use CDQ) is not enough to tackle the bias problem during stage transition. Note there are likely other methods that can achieve the same result as the Safe Q technique, and we use the Safe Q technique here because of its unique advantages: a) it is effective, and it is easy to see that it becomes difficult for Q values to diverge under Safe Q, while methods like CDQ do not provide such robostness (they do not care if the Q values are becoming too large). b) Safe Q is a minimalist solution for stage 2-3 transition, is easy to implement and has negligible overhead.

## D.2 Additional Discussion on Future Work

The success of VRL3 opens up a number of exciting future research directions in data-driven DRL.

One direction is to use on-policy methods in the VRL3 framework. Although on-policy methods tend to have worse sample efficiency due to the difficulty of using off-policy data, recent studies show that on-policy methods can use data many more times to learn useful features with value distillation [11]. In this way it might possible to combine the unique advantages of on-policy methods with the effectiveness of a data-driven approach.

When a much deeper encoder is used, VRL3 can have a larger computation cost compared to RRL and other methods that freeze their pretrained encoder. One possible idea is to use a double encoder structure, similar to [25], and to have a deeper frozen encoder and a shallow finetuning encoder. This can make the framework more complex but can potentially greatly reduce computation when using very deep encoders. Another possibility is to freeze part of the encoder and only finetune the last few layers. Currently it is unclear which will be a better solution.

Model-based methods (to list a few, [29, 5, 42, 100, 40, 10, 67, 13, 38]) can be used together with VRL3. Model-based methods tend to be more complex than model-free ones, however, these models are typically learned in a supervised manner, making them good candidates for building advanced data-driven methods. Recently, there are also model-based methods that can tackle offline RL setting [96, 32, 1]. Transformer-based methods are also an promising direction to pursue and they might provide unique advantages since they follow a supervised learning pipeline [30, 6], recently, an interesting work has shown that pretraining with data from different domains can help transformer-based RL learning [68].

Contrastive pretraining techniques can also be useful in such a data-driven approach. Although our experiments show that contrastive representation learning in stage 2 alone do not work well, they can be combined with data-augmented offline and online RL. Recently, a number of methods have shown that model-based and contrastive methods can help accelerate online learning [69, 91]. In our current design, for stage 2 and 3, we use data coming from the exact RL task we want to solve. It is also possible to utilize data from RL tasks that are related, but are not exactly the task we are solving. Here contrastive pretraining has a lot of potential.

Another important direction is to understand how the domain difference between data sources affect the learning process and how we can best utilize stage 1 data when the RL task environment has different visuals.

### D.3 Additional Discussion on Data Augmentation Technique

In our work, we use random shift data augmentation, same as our backbone algorithm DrQv2. This data augmentation technique has been shown to achieve superior performance in DRL setting, compared to other alternatives, and has been extensively studied in the DrQ paper [35]. Later on, a slightly modified version of this method is proposed to make computation more efficient [93].

In [35], section E "Image Augmentations Ablation" in the appendix, the authors provide ablation against the following alternatives: Cutout, Horizontal/Vertical Flip, Rotate, Intensity on 6 DMC tasks. Random shift achieves the best performance, and outperform the second best, Rotate, by a large margin.

### D.4 Additional Discussion on Other Recent Papers

Here we discuss how our work is related to and different from a number of recent papers (most are submitted to arxiv in 2022).

[50] show that combining self-supervised learning and RL does not necessarily leads to consistent performance improvement, this is an interesting observation and in our results we also show that self-supervised contrastive learning can be safely removed in stage 2 when offline RL is applied and updates the encoder. This work studies a different topic and is not tested on Adroit. They do not [76] show that features from pretrained networks can be used to generate semantic scene segmentation and help robotic navigation. They do not test on standard benchmarks and do no consider offline RL data. [101] show that policy pretraining on YouTube videos can help driving, they do not study offline RL and is focused on a different task. [12] study how pretrained model can be used for zero-shot goal specification, this is a difference topic and they do not test on Adroit. [57] study how transformer representations can be used for control, they do not test on the Adroit benchmark. [97] is a short paper that showed up in openreview in July, they are very similar to RRL, except they do not use demonstrations and train in an actor-critic way, they are not tested on Adroit. [71] use video prediction for pretraining, they do not consider and offline RL setting and do not test on Adroit. [79] use pretrained representations to facilitate exploration, they do not consider how to combine 3 different data sources and do not test on Adroit. [14] show that representations learned via Neural Radiance Fields can improve RL performance compared to other representations, they do not consider offline RL data and do not test on Adroit. [70] show that decoupling representation learning and dynamics learning can be helpful in model-based RL, which is different from our setting which is model-free RL. They do not have non-RL pretraining and is not test on Adroit. [9] study a more sohpisticated method on learning spatial features for visual control tasks. They do not consider the offline RL setting and do not test on Adroit. [83] study the setting where an existing policy can be used to help training. They do not study representation pretraining, and they tested on the raw-state Adroit environment, not the pixel-input one. [52] tackle robotic manipulation tasks with a sophisticated algorithm, it seems they benchmarked on raw-state Adroit setting, not the pixel-based setting. [85] study how data from multiple tasks can be used to improve performance on new tasks, the multi-task setting is different from ours, and they do not test on Adroit. [24] propose a new imitation learning method, they study a different subject, and they do not have non-RL pretraining, and do not test on Adroit. [75] show that human videos can help agents learn robotic manipulation, their subject is not exactly reinforcement learning, and they also do not benchmark on the Adroit tasks.

Note all above works are different from our work in one or more aspects, we also located a few works that have tested on the pixel-input Adroit benchmark: [65] is a thesis submitted in May, they show that performance of RRL can be slightly improved with a better pretrained representation, on Hammer, they reach 90% success rate at 3.2M data, while we are at 0.5M, so their performance is weaker than ours (we are more than 6 times more sample efficient than this result). [90] show masked image pretraining can lead to strong performance on a new suite of robotic tasks. However, they do not consider offline RL training, and do not test on Adroit. [62] and [59] show that pretraining on different datasets and with more sophisticated methods can be combined with imitation learning. Compared to them, we study RL, which is a different subject, and we also achieve better success rate on Adroit (95%, while they have 85% and <70%, respectively).

## D.5    Discussion on Potential Negative Societal Impacts

This particular work focuses on testing in simulated benchmarks, but the long-term goal is to build towards general RL methods that can achieve strong performance in challenging real-world visual control tasks.

Since our method is data-driven and utilizes large non-RL image datasets for pretraining, a potential issue is our model can be affected by the bias in these datasets. In supervised learning, there is a large body of research on how bias in the data can be harmful in many different aspects, and many of these problems can also apply to the reinforcement learning setting.

For instance, an autonomous vehicle that drives on a highway can be unsafe if its training data is unevenly distributed and certain obstacle or specific types of objects are not well represented in the data. In such cases, the agent might not be able to properly recognize these novel objects and this leads to significant safety issues. Even if the data is evenly distributed, there might be potential adversarial attacks that are designed to break data-driven intelligent systems, and special training procedures are necessary to tackle these possibilities. In a multi-stage training setting, special care should be taken on each and every training stage to ensure safety. The most popular and sample efficient DRL methods nowadays, such as SAC [23] and DrQv2 [93] do not yet have built-in procedures that aim to tackle these issues and much further research is needed to develop agents that are both efficient and reliable.