# OpenReview forum: "VRL3: A Data-Driven Framework for Visual Deep Reinforcement Learning"
_NeurIPS.cc/2022/Conference — NeurIPS 2022 Accept_

### Official Review · Reviewer_1wyj · 2022-07-10

**Rating:** 4
**Confidence:** 5
**Soundness:** 3 good
**Presentation:** 2 fair
**Contribution:** 3 good

**Summary:**

This paper proposes a 3-stage visual RL framework, consisting of encoder pretraining with ImageNet, offline finetuning, and online finetuning. The encoder pretraining stage leverages an off-the-shelf large-scale dataset to learn a powerful representation. Then, the offline pretraining uses a conservative method to transfer the learned representation into task-specific embeddings. Finally, the online rl achieves high sample efficiency in accomplishing adroit  tasks. The core contribution of this paper is the usage of imagenet dataset and offline  method to perform pretraining.  The authors also conduct detailed ablative experiments to understand the challenges and important design choices.

**Questions:**

Per the previous section, could you re-write some key sections in your paper?

Can you explain the sensitivity caused by Q threshold? If other people want to adopt this framework, how they can find a good Q threshold？

Are the data augmentations chosen randomly or on purpose? Why do you choose this augmentation?

**Limitations:**

The authors mention real robot experiment and large domain gap as their limitations. These are reasonable. Thank you for being straightforward.

**Strengths And Weaknesses:**

Strength:
Although all the components are not new, the combined framework is powerful. Within this framework, the authors conduct extensive experiments to understand possible alternatives and analyze the challenges. This shed light on future researchers who is willing to jump into this field. The sample efficiency on adroit benchmark is much higher than previous baselines and their forged stronger baselines with a large margin.

Weakness:
The overall writing quality is slightly concerning. This is an issue throughout the whole paper. For example, when the authors are describing their experimental results, they use very subjective words such as “good “, “very” without supporting evidence. In the introduction, the authors also describe the necessity of their method using the word “we want to …”. These subjective tones heavily reduce the scientific value of this paper. Moreover, there are many grammatical issues.

Slight overclaim, over-selling, and abuse of terms. The author emphasizes their method is a “minimalist” approach; a 3-stage approach with tons of tunings can hardly be regarded as minimalist. When the authors mention offline RL, they should be more careful when they are only using expert demonstrations. This needs clarification somewhere in the paper. The overall paper quality, especially in terms of the experimental aspects, is ok. That being said, I think over-selling the paper is not a good idea. The authors mentioned that it is a ResNet architecture. However, if it does not have skip connect, the naming is inaccurate.

According to the supp material, the framework is brittle to many hyperparameters. For example, the Q threshold is extremely important to tune. What would happen if other people want to use this paper? Does the tuning take a lot of effort?

The data augmentation technique seems important. More reasoning and explanation are needed.

---

> ### Author Response · Authors · 2022-08-02
> **Response to Reviewer 1wyj (3)**
>
> > The authors mentioned that it is a ResNet architecture. However, if it does not have skip connect, the naming is inaccurate.
>
> Thank you for your detailed comment, we actually did not refer to our model as a ResNet. In section 3, Stage 1, we wrote “We follow the standard training procedure of a typical ResNet model, …, In our main results, we also use a lightweight encoder with 5 convolutional layers, with batch norm after each layer”. we added a sentence to further clarify this, and also mentioned that we have new experiments on deeper ResNet architectures that do have skip connections in appendix A.7 (page 29), showing that we can further improve performance by using a higher-capacity encoder.
>
> > When the authors mention offline RL, they should be more careful when they are only using expert demonstrations. This needs clarification somewhere in the paper.
>
> Thank you for your suggestion, that is a good point! We added a sentence in section 3 to emphasize this when we introduce the Adroit task. (page 4, line 148)
>
> > The data augmentation technique seems important. More reasoning and explanation are needed.
>
> > Question: Are the data augmentations chosen randomly or on purpose? Why do you choose this augmentation?
>
> Thank you for your suggestion. The data augmentation technique we used is random shift, which has been shown (in the DrQ paper) to be superior compared to many other data augmentation techniques (rotate, cutout, horizontal/bertical flip, change in intensity) in DRL tasks, so it is chosen on purpose. This topic has been studied extensively in the DrQ and DrQv2 papers so we did not discuss it too much in our paper. An additional discussion on data augmentation is now added to Appendix D.4 (page 39).
>
> > Per the previous section, could you re-write some key sections in your paper?
>
> Of course, as discussed above, we have made several edits (changes in main text are highlighted in blue) according to your suggestions. Thank you so much for this constructive suggestion. If you have any other suggestions on the writing of our paper, please feel free to post them during the discussion period. We are happy to incorporate your suggestions and answer any further questions you may have.

---

> ### Author Response · Authors · 2022-08-02
> **Response to Reviewer 1wyj (2)**
>
> > Slight overclaim, over-selling, and abuse of terms. The author emphasizes their method is a “minimalist” approach; a 3-stage approach with tons of tunings can hardly be regarded as minimalist. When the authors mention offline RL, they should be more careful when they are only using expert demonstrations. This needs clarification somewhere in the paper. The overall paper quality, especially in terms of the experimental aspects, is ok. That being said, I think over-selling the paper is not a good idea. The authors mentioned that it is a ResNet architecture. However, if it does not have skip connect, the naming is inaccurate.
>
> > According to the supp material, the framework is brittle to many hyperparameters. For example, the Q threshold is extremely important to tune. What would happen if other people want to use this paper? Does the tuning take a lot of effort?
>
> > Can you explain the sensitivity caused by Q threshold? If other people want to adopt this framework, how they can find a good Q threshold？
>
> Thank you for your detailed comments and questions! We would like to first address the concern on whether our framework is truly simple and robust. We would like to discuss this from 2 main aspects:
>
> 1. Although the framework has 3 stages, it takes minimal effort to transition from one stage to the next:
>     * Stage 1 pretraining only needs to be done **once**. When using our method, other researchers can simply download our encoder (we release all code, data, model etc. and will provide a tutorial).
>     * we can adapt the stage 1 encoder to the stage 2 RL task with CCE in **one line of code with negligible computation overhead**.
>     * SafeQ and conservative offline update allow the **same backbone** algorithm to work well in stage 2 and 3, changing only **a few lines of code**.
>     * We show that with our framework, certain components, such as BC loss and stage 2 contrastive representation learning can be **entirely removed**, giving a simpler design with the same performance. (page 7-8, Section 5.2, 5.3)
> 2. In the paper, we present many algorithm component ablation and hyperparameter sensitivity studies. Results show there are only a few hyperparameters that are both important and sensitive, allowing other researchers to deploy this method to other tasks with minimal effort.
>     * The hyperparameter sensitivity study in appendix A.1 (page 17) shows that VRL3 is robust to most hyperparameters. Note that we use **(R)** to denote hyperparameters that are important and robust, meaning they require little or even no tuning. We use **(S)** to denote hyperparameters that are important and sensitive, meaning these will require tuning (see Figure caption on page 18).
>     * Out of the 16 hyperparameters/components studied, 9 are important to performance, and only **three** require tuning. Among these three, note that two of them already exist in the backbone algorithm (learning rate and action noise std), so **VRL3 only introduces ONE additional important and sensitive hyperparameter**, which is the encoder learning rate scale.
>     * Note that we marked the Q threshold as a "robust" hyperparameter because as discussed on page 5, line 179 of the paper ("Safe Q target technique"), we can simply **compute this threshold value**, so no tuning is required. Similarly, for many other hyperparameters, they are marked as "robust" because a wide range of values all give good results (for example, performance drops for the conservative weight only when it is set to the extreme value of 0). There are also other True/False hyperparameters (for example, whether or not to use stage 1 pretraining) that are marked as "robust" since they are simply set to True with no tuning.
>     * Based on these facts, we would like to point out that "brittle to many hyperparameters" is a **factual error**.
>
> We thank you again for your helpful comment, which helped us realize we were not emphasizing these important findings enough in our main text. We have now added more detailed discussions to the main text (page 9, line 333) to ensure better clarity for our readers.

---

> ### Author Response · Authors · 2022-08-02
> **Response to Reviewer 1wyj (1)**
>
> > Weakness: The overall writing quality is slightly concerning. This is an issue throughout the whole paper. For example, when the authors are describing their experimental results, they use very subjective words such as “good “, “very” without supporting evidence. In the introduction, the authors also describe the necessity of their method using the word “we want to …”. These subjective tones heavily reduce the scientific value of this paper. Moreover, there are many grammatical issues.
>
> Thank you very much for your detailed comments, we fully agree with you that we should strive for maximum scientific accuracy. In some parts of the paper we used terms such as “very” to express the fact that the level of improvement we see in our work is rare in the literature. To the best of our knowledge, few works in DRL can obtain an up to 1220% improvement in sample efficiency on the hardest task of a benchmark (our newest results show we can further improve to 2400% better sample efficiency by doubling the number of channels in our encoder). But we fully agree with you that such subjective tones can be harmful and we have now incorporated your suggestions into our revision.
>
> In our current revision, all subjective terms such as “very” and “highly” are entirely removed from our paper. We also emphasize our quantitative evidence, or provide a pointer to figures and numbers that support our claim, whenever possible. Here we list some of the major changes in writing:
>
> ---
> In the introduction:
>
> “We want it to be very sample efficient, and we want to make this framework as simple as possible…”
>
> Edit: We removed the subjective argument and emphasized supporting references.
>
> &rarr; “A minimalist approach is taken to make this framework as simple as possible while being sample efficient. As discussed in a number of critique papers (Deep reinforcement learning that matters by Henderson et al., Reproducibility of benchmarked deep reinforcement learning tasks for continuous control by Islam et al., A minimalist approach to offline reinforcement learning by Fujimoto et al.), simplicity in algorithm design is preferred because the performance of RL algorithms can be heavily affected by hidden details in the code, and a simpler design often leads to cleaner analysis and better reproducibility. ”
>
> ---
> “However, the environments in these benchmarks often look very different from the real world”
>
> Edit: we removed the subjective term and added a supporting reference.
>
> &rarr;  “However, the environments in these benchmarks often look different from the real world (see discussion in section 5.8, page 8 in the RRL paper)”
>
> ---
>
> “Significant improvement over the previous SOTA: VRL3 achieves an entirely new level of SOTA sample efficiency, parameter efficiency, and computation efficiency on the highly challenging Adroit benchmark while also being very competitive on DMC. ”
>
> Edit: we remove subjective terms and emphasize quantitative measures.
>
> &rarr;  “Significant improvement over the previous SOTA: VRL3 achieves a new level of SOTA sample efficiency (780\% better), parameter efficiency (3 times better), and computation efficiency (10 times faster to solve hardest task) on the challenging Adroit benchmark while also being competitive on DMC. "
>
> ---
>
> Results section:
>
> “FERM(VRL3) and DrQv2fD are both good, and VRL3 is slightly better, all three are better than DrQv2 (results from authors)”
>
> Edit: We emphasized the results can be found in Figure 2c, and removed subjective tones.
>
> &rarr; “As shown in Figure 2c, VRL3 has the best performance in the early stage, FERM(VRL3) and DrQv2fD are slightly weaker, but achieve the same performance at 1M data. All three of them outperform DrQv2 (results from authors) by 20% at 1M data”
>
> ---
>
> “Figure 3a shows that for VRL3, only using BC on expert demonstrations in stage 2 does not give a good performance. In fact, using BC in stage 2 gives a similar performance to when stage 2 is entirely disabled. “
>
> Edit: We rephrase the sentence and remove subjective terms so it is more accurate.
>
> &rarr; “Figure 3a shows that for VRL3, only using BC on expert demonstrations in stage 2 leads to the same performance compared to when stage 2 is entirely disabled. Additional ablation can be found in Appendix A.8.”
>
> ---
> Section 5 Challenges, Design Decisions, and Insights
>
> “In addition to the very strong performance on Adroit”
>
> Edit: We remove the subjective term and emphasize quantitative measures.
>
> &rarr;  “In addition to the strong performance on Adroit (compared to the previous SOTA, an average of 780% better sample efficiency, 10 times faster in wall-clock time to solve the hardest task)”
>
> ---
>
> “CCE…Such a very simple method can effectively solve…”
>
> Edit: We remove the subjective term and provide supportive evidence.
>
> &rarr; “Such a simple method (can be done in one line of code, with no additional computation overhead, ablation study against 4 other variants can be found in Appendix A.9) can effectively solve...”

---

> ### Author Response · Authors · 2022-08-07
> **Follow-up response to Reviewer 1wyj**
>
> We would like to first thank you again for your constructive comments and helpful suggestions. Since we are near the end of the discussion phase, we would like to post a follow-up response.
>
> In our previous response and our revision, we have addressed your main concern about our writing and the accuracy of claims, in summary:
>
> - we have made a number of edits according to your suggestion, we **removed all subjective terms**, and in our revision, we now further emphasize our **quantitative evidence**, or provide **a pointer to figures and numbers** that support our claim, whenever possible.
> - we clarify that "brittle to many hyperparameters" is a **factual error**. We now emphasize in the main text that the results in our appendix show that out of the 16 hyperparameters/components studied, 9 are important and **only three require tuning**. Among these three, two of them exist in the backbone algorithm (learning rate and action noise std), so VRL3 only introduces **ONE** additional important and sensitive hyperparameter (encoder learning rate scale).
> - We also clarified that the Q threshold is **computed** and does not require tuning.
> - A number of other modifications are made to further refine the paper. Additional discussion on data augmentation is added to Appendix D.4 (page 39). Experiments with deeper and wider encoders are added to Appendix A.7 (page 29), showing a higher capacity encoder can further improve performance on Relocate, solving it 24 times faster than RRL, etc.
>
> Do you find this response satisfactory? Or if you have any other suggestions on further improving the writing, it would be great if you can post them during the discussion phase, so we can add them to our paper and submit a new revision before the discussion phase ends. Thank you very much for your time and effort in providing valuable feedback on our work!

---

> > ### Comment · Reviewer_1wyj · 2022-08-07
> > **Response to the rebuttal**
> >
> > The authors address some of my concerns. However, I still think the paper is below the bar.
> > 1. The authors improve a lot on paper writing. But submitting a drafty manuscript, then heavily revising during rebuttal is not the best way to get a paper into NeurIPS.
> > 2. The minimalist claim is still too strong for me. While between stages, it requires only a few changes, the overall modification is not minimal. The authors argue that other researchers can download the pre-trained encoders. However, this is not sufficient to support that their approach is minimalist. For example, people can download alphago/resnet/SAC and run it. But would you call all of them the minimalist method?
> > 3. Q thresh robustness. The author mentioned in the paper that the Q thresh can be computed or set a bit higher than the value. But in the appendix, when the Q threshold is higher, the performance dramatically drops.
> > 4. I think the reviewers have already made some improvements. But due to the previous concerns and thoughts, I cannot raise the score.

---

> > > ### Author Response · Authors · 2022-08-07
> > > **Further response to Reviewer 1wyj**
> > >
> > > > The authors improve a lot on paper writing. But submitting a drafty manuscript, then heavily revising during rebuttal is not the best way to get a paper into NeurIPS.
> > >
> > > Thank you for your comments. We would like to point out that although we made changes in the writing, the core contributions of the paper **never changed**. We had all the content already when we submit the paper, and if you look at the edits we made, all we are doing is further emphasizing the evidence in our paper. And as we stated before, in the original version, for example, we used the term "very good" to describe our performance gains, because they are indeed very good. We have an average of 780% improvement in sample efficiency, and we can solve the hardest task 24 times faster than previous SOTA, this is not a number **often seen in the literature**.
> > >
> > > We would also like to point out that we are now having a discussion phase and revisions for this conference because we would like research papers to be improved during the reviewing process, and we very much appreciate your effort in helping us improve the paper. Now the paper has been improved, shouldn't we focus on **the improved paper**, instead of ignoring all the effort made to improve it?
> > >
> > > > The minimalist claim is still too strong for me. While between stages, it requires only a few changes, the overall modification is not minimal. The authors argue that other researchers can download the pre-trained encoders. However, this is not sufficient to support that their approach is minimalist. For example, people can download alphago/resnet/SAC and run it. But would you call all of them the minimalist method?
> > >
> > > We call it a "minimalist approach" because it is **the minimal way to effectively combine data from all three stages**. If we compare to a method that cannot utilize data from 3 stages, for example, a pure online method, then, of course, we are not "minimal". But if we compare to **other alternatives to utilize data from these 3 stages**, then we are indeed the minimal approach. For example, in the paper we have shown that BC can be entirely removed, and additional stage 2 contrastive learning can be removed, these are potential additional components that can be added to the framework, and they are removed to leave only the most critical ones.
> > >
> > > > Q thresh robustness. The author mentioned in the paper that the Q thresh can be computed or set a bit higher than the value. But in the appendix, when the Q threshold is higher, the performance dramatically drops.
> > >
> > > We have to point out that it only happens at an **extreme value** of 1000 (10 times higher than the computed value, this is definitely not "a bit higher"). Note that near the computed value of 100, performance is always good. We report extreme values because this is **the correct way to perform a hyperparameter sensitivity study**, when an extreme value is applied, then we **should** see a difference in performance, even though the hyperparameter is robust. In this particular case, setting it to an extremely high value is the same as not using this technique at all, so we expect to see a performance drop. This does not change the fact that we can directly compute this hyperparameter, and it does not require tuning.
> > >
> > > > I think the reviewers have already made some improvements. But due to the previous concerns and thoughts, I cannot raise the score.
> > >
> > > The use of terms and grammar is important, however, we encourage the reviewer to evaluate our work based on our actual contributions, such as our novel framework, our significant performance gain, our extensive ablation experiments, and the many insights we discovered during our research. We are a team of researchers from multiple cultural and language backgrounds. The way we use language might not be the most orthodox, but we believe the contributions in this paper are indeed significant and can benefit the research community.

---

> > > ### Author Response · Authors · 2022-08-09
> > > **A more detailed explanation of why we believe VRL3 has a "minimal" design**
> > >
> > > We would like to thank you again for your comments and questions, we would like to provide a more detailed (and a more visual) explanation of why we believe our framework has a "minimal design". We hope this will make things more clear. In the table below, **The 3 BOLD ROWS** combines to VRL3. All other rows are potential alternatives that we investigated and then discarded either due to lower performance, higher computation overhead, or higher implementation complexity.
> > >
> > > This table shows why we call the final design for VRL3 a "minimal" design. VRL3 can fully exploit data from 3 stages to achieve strong performance while being the fastest to run and most simple to implement. (NOTE that we are, of course, not saying we are more minimal than a pure online method. The goal is, to find **the minimal design that can fully exploit data from all 3 stages and achieve superior performance**. So we are comparing alternative design choices that can exploit data from 3 stages. We are not considering methods that cannot exploit data from 3 stages.)
> > >
> > > | Design choice     | Improve performance | Additional computation overhead | Implementation complexity (how much it deviates from our DrQv2 backbone) | Evidence/Discussion |
> > > | ----------- | ------|----- | ---| --- |
> > > | S1 Naive   | No | Negligible  | No modification  | P7 Fig 3
> > > | **VRL3: S1 CCE**   | **Yes**  | **Negligible**  | **One line of code**  | P31 Fig 20
> > > | S1 Random First Layer   | No  | Negligible  | One line of code|  P31 Fig 20
> > > | S1 Encoding Sum |No | Up to 4 times slower for all updates   |  A few lines of code |  P31 Fig 20
> > > | S1 Encoding Cat  |  Yes  | Up to 4 times slower for all updates      | A few lines of code  |  P31 Fig 20
> > > | S2 naive | No | A fixed amount of updates  | No modification |P7 Fig 3b, 3c
> > > | S2 naive + Safe Q | No | A fixed amount of updates  | A few lines of code | P7 Fig 3
> > > | **VRL3: S2 conservative** | **Yes** | **A fixed amount of updates**  | **A few lines of code** |P6 Fig 2
> > > | S2 additional BC |  No | A fixed amount of updates  | A few lines of code  | P18 Fig 6h
> > > | S2 BC with naive value learning |  No | A fixed amount of updates  | More coding  | P30 Fig 19
> > > | S2 additional contrastive | No | A fixed amount of updates  | More coding | P6 Fig 2a, 2b
> > > | S2 other more sophisticated offline methods | Unclear | Depends  | Deviate significantly from backbone |P3 Sec 2.3
> > > | S3 naive | No | Negligible  | No modification |P7 Fig 3b, 3c
> > > | **VRL3: S3 safe Q** | **Yes** | **Negligible**  | **A few lines of code** | P7 Fig 3b, 3c
> > > | S3 conservative | No | Fixed amount of additional computation per update | A few lines of code | P7 Fig 3b, 3c
> > > | S3 additional BC updates | No | Fixed amount of additional computation per update  | A few lines of code | P18 Fig 6n, 6o
> > > | S3 other more sophisticated offline-online methods | Unclear | Depends  | Deviate significantly from backbone | P3 Sec 2.3

---

> > > ### Author Response · Authors · 2022-08-09
> > > **A more detailed explanation of hyperparameter importance and robustness**
> > >
> > > We would like to thank you again for your comments and questions, and we would like to also post a more detailed explanation of how the importance and robustness of a hyperparameter are decided in our study, just to make things more clear.
> > >
> > > In our hyperparameter study, we put each of the 16 algorithmic components/hyperparameters into one of the three categories:
> > >
> > > - **Not important**
> > > - **Important and robust (R)**
> > > - **Important and sensitive (S)**
> > >
> > > Note that, when all values of a hyperparameter lead to the same result (even for extreme values), we do not call it **robust**, instead, we simply say this hyperparameter is **not important**. For example, when we change the BC loss weight, all values from 0 to 1 give essentially the same performance. This shows that whether BC is enabled or not, the performance is the same. So BC can be entirely discarded because its contribution is negligible. 7 out of the 16 hyperparameters/components are categorized as "not important" due to this reason.
> > >
> > > For hyperparameters that **actually affect performance**, if they are set with a boolean of True/False (e.g. enable data augmentation), or if good performance can be achieved with a wide range of values, then they are considered **important and robust** because they require minimal or even no tuning. Note that Safe Q threshold belongs to this category because only extremely high values (10 times the computed threshold) lead to a performance drop. When setting a very high threshold, we are essentially removing this regularization entirely, so a performance drop is expected. 6 out of 16 belong to this category.
> > >
> > > For hyperparameters that affect performance, if non-extreme values can lead to very different performance, then they are considered **important and sensitive**. For example, unsurprisingly, learning rate and action noise std belong to this category (learning rate requires tuning for most DL algorithms, and action noise has been shown to be important in most DRL algorithms). 3 out of the 16 belong to this category. And these 3 are the ones that actually **require tuning**. So in the end, VRL3 only introduces **ONE** additional hyperparameter that is **important and sensitive**: the encoder learning rate scale.

---

### Official Review · Reviewer_vuXw · 2022-07-11

**Rating:** 6
**Confidence:** 4
**Soundness:** 3 good
**Presentation:** 4 excellent
**Contribution:** 3 good

**Summary:**

The authors propose VRL3, a 3 stage data driven technique for pretraining convolutional encoders, performing offline RL and fine tuning with online RL. The authors demonstrate their approach in a number of vision based control benchmarks, include Adroit and DMC.
The approach greatly improved sample efficiency and outperforms end-to-end methods.

The usage of offline datasets and pretrained encoders if an interesting field of work, but I am concerned that the problems this framework is evaluated on are somewhat trivial and I would be interested to see the results after more iterations of fine tuning.

**Questions:**

* The authors retrain their autoencoder on 84x84 images, given they are using the convoluational part, why could they not take the weights from an already existing pretrained resnet 18, for example?
* What is the benefit of a more convoluted method if I can train my baseline for more iterations and achieve comparable performance?
* How applicable is this process to other RL algorithms, could you have made comparisons to on-policy training and memory-based agents?

**Limitations:**

Yes

**Strengths And Weaknesses:**

Strengths:
* The paper and methodolgy is clear and easy to read.
* Improved sample efficiency.
* The authors share their codebase and plan to release it as open source, with the data, checkpoints etc.
* This method, could potentially be applied to a large variety of benchmarks and achieve SOTA in terms of sample efficiency.

Weaknesses:
* The method requires training an encoder at the observation size of the environment.
* I am not convinced that this approach outperforms pure online methods if they are left to convergence, figure 2 (a) show the performances of RRL still increasing, has this baseline actually converged
* I am not convinced the approach would transfer well to POMDP settings and more challenging benchmarks (DeepMind Lab, Habitat-lab ...)
* It would have been interesting to see comparisons of other pretraining regimes for the encoder, (self-supervised, BOYL, contrastive.. etc)

---

> ### Author Response · Authors · 2022-08-02
> **Response to Reviewer vuXw (2)**
>
> > Question: What is the benefit of a more convoluted method if I can train my baseline for more iterations and achieve comparable performance?
>
> Thank you for this excellent question. Here are the reasons:
>
> - Better **sample efficiency** (also discussed above). compared to RRL, we are on average 780% faster to reach 90% success rate in all tasks. In appendix B.1 (page 32) we provide details on how we compute these numbers and also provide comparison on how fast we are to reach 50% and 75% success rate. In all cases, VRL3 has a significant advantage, allowing DRL training to be **cheaper**.
> - Better **computation efficiency**, meaning a **faster** wall-clock time. VRL3 is 10 times faster in solving the most challenging Relocate task. So for example, when a researcher is using the prior SOTA method, experiments might finish after 2 days, but when using our method, you will be able to see the results in 5 hours. This is especially beneficial for researchers in small labs with limited computation power.
> - Better **parameter efficiency**, compared to the previous SOTA, our encoder is 50 times smaller and the overall agent is 3 times smaller. This means we require a smaller amount of memory when deployed on actual robotic systems, making it more **portable**.
> - Our extensive ablations show that VRL3 is actually not very “convoluted”. For example, although it has 3 stages, stage 1 training only needs to be **done once**, and the transition between stages is easy (CCE allows transition to stage 2 with **one line of code with no overhead**, SafeQ allows the **same backbone** algorithm to work well in stage 2, 3, adding only **a few lines of code**). The hyperparameter sensitivity study in appendix A.1 (page 17) further shows that VRL3 is robust to most hyperparameters, in fact, out of the 16 hyperparameters/components studied, 9 are important to performance, and only **three** require tuning. And if we ignore the hyperparameters that already exist in the backbone algorithm (learning rate and action noise std), we can see that VRL3 introduces **only one** important and sensitive hyperparameter, which is the encoder learning rate scale. In summary, the design of VRL3 allows us to take all the above advantages with minimal effort.
>
> > I am not convinced the approach would transfer well to POMDP settings and more challenging benchmarks (DeepMind Lab, Habitat-lab ...)
>
> That is a valid point. Although Adroit from pixels can be seen as a POMDP task (because information not captured by the camera is not available to the agent), the fact that we achieved great success on Adroit does not automatically prove we can also achieve SOTA performance on more challenging benchmarks. That is why we only claimed SOTA on Adroit. We believe our work takes an important step toward better data-driven DRL algorithms, but certainly will not be the last paper in this direction. We will look into these other benchmarks and add a discussion to our revision.
>
> > It would have been interesting to see comparisons of other pretraining regimes for the encoder, (self-supervised, BOYL, contrastive.. etc)
>
> That is a good point, in section 5.2 and 5.3 of our paper (page 7-8), our results show that during stage 2 training, offline RL training can be superior to just perform contrastive representation learning (FERM uses contrastive learning in stage 2, Figure 2 (page 6) shows FERM+stage 1 pretraining is still weaker than VRL3). This might be due to the fact that contrastive representation learning
> only trains the encoder and not the policy and the value networks.
>
> In terms of different pretraining methods for stage 1, we are working on new experiments now and will add the results to the paper when they are ready.
>
> > How applicable is this process to other RL algorithms, could you have made comparisons to on-policy training and memory-based agents?
>
> Thank you for your question, the RRL algorithm is essentially an on-policy method, and based on our results, we believe that VRL3 (with an off-policy algorithm backbone) has an advantage over on-policy methods because it is easier for off-policy methods to fully exploit offline and online data.
>
> For memory-based agents, they mainly leverage an external memory to help policy learning, and it is non-trivial to design memory-based algorithms to well utilize non-RL, offline RL and online RL data with external memory, and this can form another new interesting work.
>
> The promising results in our work demonstrated how leveraging all available data sources can greatly boost DRL performance, and we believe such a data-driven mindset can also be helpful for memory-based methods, and we leave this as our future work.

---

> ### Author Response · Authors · 2022-08-02
> **Response to Reviewer vuXw (1)**
>
> > The method requires training an encoder at the observation size of the environment.
>
> > Question: The authors retrain their autoencoder on 84x84 images, given they are using the convoluational part, why could they not take the weights from an already existing pretrained resnet 18, for example?
>
> Thank you for your comment and question. Training an encoder at the observation size is NOT a hard requirement of our method, and yes we can totally take a pretrained ResNet 18. We did that in the beginning stage of our research, but later we decided to retrain a smaller stage 1 encoder on 84x84 for the following reasons:
>
> - An input size of 84x84 with a shallow encoder are used by many popular pixel-input DRL methods (for example, DQN, RAD, DrQ, DrQv2, etc.), and we want to be more consistent with these prior works.
> - This decision makes computation much faster, which is preferred when large amounts of experiments are required.
> - Our results show that 84x84 input with a shallow encoder already achieves superior performance. (We have a new result showing that we can boost performance even further with a higher capacity encoder)
> - Stage 1 pretraining only needs to be done once. When using our method, other researchers can simply download our encoder (we release all code, data, model etc. and will provide a tutorial).
>
> Additionally, we now have new results in appendix A.7 (page 29) showing that VRL3 can also train with deeper and wider ResNet encoders (ResNet6 with 64 channels, ResNet10, ResNet18 with 32 and 64 channels) with default hyperparameters. It is worth mentioning that with ResNet6 and 64 channels, we achieve even better performance: VRL3 is able to reach 90% success rate on the hardest Relocate task with only 0.45M data, making it 24 times more sample efficient compared to the previous SOTA.
>
> > I am not convinced that this approach outperforms pure online methods if they are left to convergence, figure 2 (a) show the performances of RRL still increasing, has this baseline actually converged
>
> We now added figures and discussion on long-term performance of RRL in appendix A.6 (page 28). Results (from RRL authors) show that when RRL is trained to 12M data (3 times the data usage of VRL3), its long-term performance is similar to VRL3, but still a bit lower. VRL3 is slightly stronger in Relocate, and in Pen, VRL3 reaches 90% success rate at 3.25M data and then continues to increase, RRL fails to reach 90% success rate even at 12M data.
>
> Based on these numbers, it is fair to say that VRL3 outperforms RRL in both early stage and late stage.
>
> If RRL is trained for even more data points, it is reasonable to assume it can eventually reach the same performance. Although the long-term performance gap is small, the difference in sample efficiency is significant. That is why in the paper we mainly focused on discussing the improvement in **sample efficiency**, a commonly used measure in DRL research: if an algorithm can achieve a certain level of performance using only a small amount of online RL data, then it has a high sample efficiency.
>
> It is important because in the real world, collecting online RL data (for example, with a sophisticated robotic system) can be risky and expensive. So in many research papers, we assume that computation is relatively cheap, and online data is expensive. For example, if a company is training a robotic system similar to Adroit, and each online data point (by actually running the robot to interact with the real world) costs 1 dollar, then RRL will cost 11 million dollars to learn Relocate, while VRL3 only costs 0.9M dollar (we are only making an example here to deliver the idea, the actual cost depends on the robotic system).

---

### Official Review · Reviewer_5Wwb · 2022-07-14

**Rating:** 6
**Confidence:** 3
**Soundness:** 3 good
**Presentation:** 3 good
**Contribution:** 3 good

**Summary:**

The authors tackle the problem of improving the efficiency of online learning for visual RL tasks. The main idea of the paper is to pre-train an encoder on non-RL data (Imagenet), then train a value function and policy network on top of the shared encoder with offline RL, and finally fine-tune with online data. With this simple framework and a small number of technical tweaks (e.g. how to pre-train the encoder with single-image input but transfer to tasks with multi-image input, how to stabilize the value function), the authors show that their method achieves SOTA sample efficiency on the Adroit benchmark.

**Questions:**

- Lines 66-68 - I missed if the authors ablate CCE? How does performance compare if the first layer is simply randomly initialized for stage 2?
- Lines 249-251 - a 5-layer CNN seems quite limiting in terms of the range of tasks that such a network can solve. What's the Imagenet accuracy of this network after Stage 1 pre-training? Is the framework the authors propose stable for deeper networks?
- Figure 2 - these curves would be easier to read if the legend was moved outside the plot. I also recommend reordering the labels in the legend to match the ordering of the curves in the graph.
- Algorithm 1 - For me at least I found the algorithm box too confusing to be helpful for understanding the method. I personally feel it distracts from the clarity of description in Section 3 and Figure 1.
- Lines 173 - 177 - "To allow more flexibility in the Q network, we can also set the threshold to be higher than this value". Can the authors elaborate more on why it makes sense to allow the Q network to output larger values than what is already a very high upper bound on the range of Q? I'm also wondering why the authors don't parameterize Q with tanh or sigmoid in order to enforce the correct range and instead use the technique they describe in line 176.
- Line 307 - "to" I believe should be "than"

**Limitations:**

Yes - the authors discuss that experiments on simulated benchmarks do not imply success in the real world.

**Strengths And Weaknesses:**

I found this paper a breeze to read. The method is very practical and leverages popular tools in the contemporary ML toolbox (pre-training and fine-tuning, offline RL) in order to achieve impressive sample efficiency on synthetic visual-input robotic tasks. I especially appreciate the careful ablations the authors performed in Section 5 which I believe gives insight into the contribution of each of the 3 stages that the authors propose for their method. I think this paper provides valuable insight for the ML community on the relative importance of leveraging non-RL, offline, and online data for RL tasks.

The main weakness I see for this paper is I think the authors make some claims about the importance of offline RL vs. BC in section 5.2 that I think need an additional experiment to verify. In lines 263-265, the authors claim that BC is the same as entirely ignoring stage 2 altogether "since BC only trains the actor and not the critic". I think BC should provide a stronger baseline than this - maybe the authors could try a stage 2 in which the policy is trained with BC and the value function is trained in a naive way to predict cumulative reward of the offline trajectories? My takeaway from Figure 4 is that as long as we have a decent policy-value match going into stage 3, the tasks will get solved even if the encoder hasn't even been trained yet (with smoother more efficient learning when the encoder is pre-trained on Imagenet). It would be great to disentangle the intricacies of DrQv2 for offline data from the goal of pre-training both the policy and value function in stage 2.

---

> ### Author Response · Authors · 2022-08-02
> **Response to Reviewer 5Wwb (2)**
>
> > Lines 249-251 - a 5-layer CNN seems quite limiting in terms of the range of tasks that such a network can solve. What's the Imagenet accuracy of this network after Stage 1 pre-training? Is the framework the authors propose stable for deeper networks?
>
> Thank you for this very good question, indeed this 5-layer CNN (We refer to this default encoder as ResNet6 with 32 channels) is quite small with limited capacity, we have conducted additional experiments using deeper and wider ResNet models, below is a list of models we tested and their ImageNet validation accuracy:
>
> | Encoder Name     | # Conv Layers| ImageNet Top-1 Acc |Top-5 Acc|
> | ----------- | ------|----- | ---|
> | ResNet6 32Channel   | 5  | 34.7       | 58.6   |
> |  ResNet6 64Channel  |5 | 43.7        |  68.3  |
> | ResNet10 32Channel  | 9   | 48.6      | 72.8  |
> |  ResNet10 64Channel | 9 | 59.7       | 81.8   |
> | ResNet18 32Channel  |  17  | 57.6       |80.8    |
> |  ResNet18 64Channel  | 17| 66.6        | 87.1   |
>
> On the hardest Relocate task, we found that when using the default hyperparameters, their performances are similar, while ResNet6 with 64 channel achieves the best sample efficiency, **even better** than the results originally reported in the paper (it reaches 90% success on Relocate just after 0.45M data, making it 24 times more sample efficient compared to RRL). With default hyperparameters, we observe weaker performance on the deeper ResNet18. Since these networks have much higher capacity, the learning rate might need to be finetuned for the best performance when using deeper networks.
>
> For the majority of our paper, we focus on the ResNet6 with 32 channel for the following reasons:
> - Such shallow encoders are used by many popular pixel-input DRL methods (for example, DQN, RAD, DrQ, DrQv2, etc.), and we want to be more consistent with these prior works.
> - Smaller models lead to faster experimentation, which is preferred since we have a large amount of experiments to run.
> - Our results show that in our framework, a shallow encoder already enables superior performance compared to the previous SOTA.
>
> > Figure 2 - these curves would be easier to read if the legend was moved outside the plot. I also recommend reordering the labels in the legend to match the ordering of the curves in the graph.
>
> Thank you for your suggestion! We will modify the figures accordingly.
>
> > Algorithm 1 - For me at least I found the algorithm box too confusing to be helpful for understanding the method. I personally feel it distracts from the clarity of description in Section 3 and Figure 1.
>
> That is a good point! Algorithm 1 is now moved to appendix C.5 for better clarity in the main paper.
>
> > Lines 173 - 177 - "To allow more flexibility in the Q network, we can also set the threshold to be higher than this value". Can the authors elaborate more on why it makes sense to allow the Q network to output larger values than what is already a very high upper bound on the range of Q?
>
> Thank you for pointing this out, intuitively, we consider the Q threshold a regularization, so we might want to only apply a minimal amount of regularization to reduce interference with the learning process. Empirical results show that setting a slightly lower or higher threshold than the computed threshold still provides strong performance (as shown in Figure 1e in Appendix A (page 18), the computed threshold value is 100, and a value of 50 or 200 also provides similar performance), showing that the SafeQ technique is quite robust. It seems that as long as the Q values do not grow too large, it is possible for the Q network to “recover” to reasonable values. We have modified the text to emphasize that this comes from our hyperparameter ablation results.
>
> > I'm also wondering why the authors don't parameterize Q with tanh or sigmoid in order to enforce the correct range and instead use the technique they describe in line 176.
>
> We don’t parameterize Q with tanh or sigmoid because although it feels like a natural solution, this has the potential problem of activation **saturation**. Take tanh as an example, if we call the value that goes into the tanh function the “pretanh” value, then what might happen during training is: it is possible that at some point in the training process, the pretanh value becomes quite high due to accumulation of Q bias, now although the Q value will not be higher than 1 because of the tanh function, the pretanh value might be too big that gradient starts to vanish through the tanh function (this issue has been discussed in the literature, for example “Deep reinforcement learning in parameterized action space” by Hausknecht and Stone). In such a scenario, the Q value might “get stuck” at a value of 1 and cannot be reduced even if the correct Q value should be lower.
>
> >Line 307 - "to" I believe should be "than"
>
> Fixed!

---

> > ### Comment · Reviewer_5Wwb · 2022-08-08
> > **Response**
> >
> > Thank you for the clarifications and ablations! I think this paper is solid and deserves to be at NeurIPS. I agree with the other reviewers that the framework is not particularly novel and the CCE and Q thresholding are not particularly innovative, but I find the simplicity of the 3 components and the experimental results compelling. I think 6 is an appropriate score for this paper and encourage the other reviewers to raise their scores.

---

> ### Author Response · Authors · 2022-08-02
> **Response to Reviewer 5Wwb (1)**
>
> > The main weakness I see for this paper is I think the authors make some claims about the importance of offline RL vs. BC in section 5.2 that I think need an additional experiment to verify. In lines 263-265, the authors claim that BC is the same as entirely ignoring stage 2 altogether "since BC only trains the actor and not the critic". I think BC should provide a stronger baseline than this - maybe the authors could try a stage 2 in which the policy is trained with BC and the value function is trained in a naive way to predict cumulative reward of the offline trajectories? My takeaway from Figure 4 is that as long as we have a decent policy-value match going into stage 3, the tasks will get solved even if the encoder hasn't even been trained yet (with smoother more efficient learning when the encoder is pre-trained on Imagenet). It would be great to disentangle the intricacies of DrQv2 for offline data from the goal of pre-training both the policy and value function in stage 2.
>
> Thank you for your helpful comment and suggestion, this is a very good point! We conducted experiments on 2 new BC variants:
>
> a) In stage 2, we perform BC updates, and we also train the value network to predict discounted cumulative reward of the offline trajectories (we call this “BC MC”).
>
> b) we train the value network with a slightly different method, we perform the standard Q-learning updates, but when computing the Q target, we replace the “next action from current policy” with the actual next actions in the offline trajectories, so as to circumvent the extrapolation error issue (we call this “BC Offline Act”).
>
> We then compare these 2 new variants with VRL3 (with offline RL in stage 2) and BC naive (the original BC baseline that only learns policy). The results are presented in appendix A.8 (page 30).
>
> Before we saw the results, we expected that these two new baselines would obtain better performance than the BC naive. However, the result is quite interesting: First we observe that BC Offline Act achieves better performance than BC MC (which is not surprising because we expect BC Offline Act to learn a better value function since it uses actions in the offline trajectory), but then we also observe that both new variants achieve **weaker** performance compared to BC Naive.
>
> We did not expect this result because as you mentioned in the comment, intuitively we would think that a somewhat decent policy-value match should certainly outperform a policy with untrained value function.
>
> We pondered on this new result, and a potential explanation is that if we train the value network in a naive way in stage 2, this might inject harmful inductive bias into the value network, making it hard to perform reinforcement learning in the later stage. There have been a few very recent papers that discuss this issue: for example, in “The Primacy Bias in Deep Reinforcement Learning”, Nikishin et al. show that early stage training, especially on a small dataset can impose a permanent effect on RL agents and reduce their long-term performance. Then it can be possible that an untrained Q network learns better in stage 3 compared to a naively trained Q network, because it is free from the inductive bias generated by the value learning process in BC MC and BC Offline Act.
>
> This is a very interesting finding, a discussion is now added to appendix A.8 (page 30).
>
> > Lines 66-68 - I missed if the authors ablate CCE? How does performance compare if the first layer is simply randomly initialized for stage 2?
>
> We now performed 3 sets of ablations for CCE: 1) We initialize the first encoder layer randomly (Rand First). 2) We treat each input image separately, and then sum their encoding (Encoding Sum) 3) Similar to 2), but we concatenate their encoding (Encoding Concat). The preliminary results are presented in appendix A.9 (page 31).
>
> Results show that Rand First and Encoding Sum have a significant performance drop in Pen and Relocate. Encoding Concat has a performance drop in Hammer and Pen, but is similar to CCE on Door and Relocate. Additionally, Encoding Sum and Encoding Concat cause an extra computation overhead, making the experiments run up to 4 times slower in Relocate.
>
> Based on these results, it seems CCE is still the best fit for our framework, because it has strong performance, is easy to use (can be done in one line of code), and has no computation overhead.

---

### Official Review · Reviewer_i5GH · 2022-07-18

**Rating:** 3
**Confidence:** 4
**Soundness:** 2 fair
**Presentation:** 2 fair
**Contribution:** 2 fair

**Summary:**

This paper outlines a 3-stage process to train policies. Stage 1 involves training an encoder using large image datasets to obtain task-agnostic visual representation. Stage 2 involves training a policy using offline RL and the pre-trained encoder. This serves a as a good initialization for the final stage 3 that involves further training the agent with online RL.

**Questions:**

Please see strengths and weaknesses section for details and context. My main questions/concerns are related to:

1. Novelty of VRL3 and its relationship to prior work (that appeared sufficiently before NeurIPS deadline) -- MVP, PVR, R3M etc.

2. Lack of rigorous comparisons to substantiate utility of CCE and SafeQ over current SOTA baselines (as opposed to naive straw-man baselines).

**Limitations:**

The limitation section in the paper is too generic (e.g. evaluation in sim vs real robots).

**Strengths And Weaknesses:**

Strengths: The overall pipeline appears obvious and straightforward. In my view, the main contribution of the paper is putting together the components and demonstrating empirical success.

Weaknesses: While the overall approach and results look promising, the current submission unfortunately lacks technical depth and rigor. In the introduction section, the authors outline their main contributions. Unfortunately, I do not believe there is sufficient evidence to rigorously substantiate these claims.

1. "Novel framework..." -- I believe there is limited novelty in this submission, especially in light of several recent papers (most of which are published by now). For example, the current submission does not cite or discuss MVP[1], PVR[2], R3M[3] which all explore training encoders from out of domain data.

2. "Novel technical contributions..." the main technical contribution claims are: (a) CCE or convolutional channel expansion; (b) safe Q technique. As for CCE, there are no experiments to benchmark it against simple alternatives like contatenation of framewise embeddings [1,2,3] or their latent differences [4], which have been successfully used in several prior works. The safeQ update does seem to help compared to a naive update rule. However, the challenges in transitioning from offline RL to online RL has been well-documented in prior work, along with different solutions (e.g. AWAC [5]). The authors should consider comparisons with such SOTA baselines as opposed to naive straw-man baselines. Overall, due to the lack of rigor in comparisons and baselines, it is hard to judge the importance or utility of the claimed technical contributions.

"VRL3 achieves an entirely new level of SOTA sample efficiency, parameter efficiency, and computation efficiency on the highly challenging Adroit benchmark while also being very competitive on DMC." -- I would encourage the authors to rephrase such statements to more technically rigorous and falsifiable claims.

[1] Xiao et al. Masked Visual Pre-training for Motor Control. arXiv 2022.

[2] Parisi et al. The Unsurprising Effectiveness of Pre-Trained Vision Models for Control. ICML 2022.

[3] Nair et al. R3M: A Universal Visual Representation for Robot Manipulation. arXiv 2022.

[4] Shang et al. Reinforcement Learning with Latent Flow. NeurIPS 2021.

[5] Nair et al. AWAC: Accelerating Online Reinforcement Learning with Offline Datasets. 2020.

---

> ### Author Response · Authors · 2022-08-02
> **Response to Reviewer i5GH (3)**
>
> > "VRL3 achieves an entirely new level of SOTA sample efficiency, parameter efficiency, and computation efficiency on the highly challenging Adroit benchmark while also being very competitive on DMC." -- I would encourage the authors to rephrase such statements to more technically rigorous and falsifiable claims.
>
> Thank you for your kind suggestion! After looking carefully at the papers you mentioned, we also searched online and identified **another 17 papers** (most of which have been released online in 2022) that might be related to our work to some extent. In appendix D.5 (page 40) we discuss how our work is different from all these works and show that our contributions are indeed unique and novel. None of these works study the subject of combining non-RL data, offline RL, and online RL data in a simple framework. And none of them show a better performance on Adroit (from pixel input). In a thesis that was submitted in May, it has been shown that for RRL, the performance on Hammer can be improved with a better representation, achieving a 90% success rate at 3.2M data. However, we can achieve this with just 0.5M data, outperforming it by a large margin.
>
> Based on these results, it seems that it is still valid to say that our work has made a number of important contributions, and achieved a new SOTA on the Adroit benchmark (an average of 780% better sample efficiency and solves the hardest Relocate task 10 times faster). In fact, after incorporating the suggestion by reviewer 5Wwb, we found that we can now solve Relocate 24 times faster than the previous SOTA by simply using a higher-capacity encoder. To further address your concern and the concern of other reviewers, we also went through the paper carefully and made a few edits to our writing to make them more scientifically accurate.
>
> If you have any other questions or concerns, or if you are aware of any other recent papers that we should cite, please feel free to let us know during the discussion period. We appreciate your time and effort, and we are more than happy to incorporate your suggestions to make our work a better one.
>
> > The limitation section in the paper is too generic (e.g. evaluation in sim vs real robots).
>
> Thank you for your suggestion, please note that in the limitation section, we also mentioned it remains unclear whether stage 1 training is still beneficial if the domain gap between non-RL data and RL data is too great (RL task visuals are different from the real world). This is a limitation specific to multi-stage frameworks that utilize data from a different source for pretraining. And this is an important question that requires further research and investigation. We have now also revised our limitation discussion and we also pointed out that we focused on experimenting with model-free methods, so it is unclear whether model-based methods can also work well with our framework, and further research is required.

---

> ### Author Response · Authors · 2022-08-02
> **Response to Reviewer i5GH (2)**
>
> > "Novel technical contributions..." the main technical contribution claims are: (a) CCE or convolutional channel expansion; (b) safe Q technique. As for CCE, there are no experiments to benchmark it against simple alternatives like contatenation of framewise embeddings [1,2,3] or their latent differences [4], which have been successfully used in several prior works.
>
> That is a good point. In appendix A.9 (page 31), we present ablation on CCE and compare it to three variants:
> 1) We initialize the first encoder layer randomly (Rand First).
> 2) We treat each input image separately, and then sum their encoding (Encoding Sum)
> 3) Similar to 2), but we concatenate their encoding (Encoding Concat). Results show that Rand First and Encoding Sum have a significant performance drop in Pen and Relocate. Encoding Concat has a performance drop in Hammer and Pen, but is similar to CCE on Door and Relocate.
>
> Another baseline is to concatenate the differences of image encodings, according to the paper “reinforcement learning with latent flow” (Encoding Diff Concat). Note that this variant, together with Encoding Sum and Encoding Concat will incur additional computation overhead because the minibatch becomes much larger going into the encoder. On a V100 GPU, for the Relocate task, 1000 updates for CCE/Rand First take about 47 seconds, while for Encoding Sum, Encoding Concat and Encoding Diff Concat, they take about 200 seconds, about **4 times slower** than using CCE. We are running experiments on this variant now and will update the figures when the experiments finish.
>
> Based on our preliminary results, it seems CCE is still the best fit for our framework, because it has **strong performance**, is easy to use (can be done in **one line of code**), and has **no computation overhead**.
>
> > The safeQ update does seem to help compared to a naive update rule. However, the challenges in transitioning from offline RL to online RL has been well-documented in prior work, along with different solutions (e.g. AWAC [5]). The authors should consider comparisons with such SOTA baselines as opposed to naive straw-man baselines. Overall, due to the lack of rigor in comparisons and baselines, it is hard to judge the importance or utility of the claimed technical contributions.
>
> Thank you for the comments, we would like to first point out that AWAC is, in fact, **not a SOTA** on Adroit with pixel input. We understand that this can be a little confusing: AWAC is actually a SOTA on Adroit with proprioceptive input (the so-called “raw-state” input). But AWAC is not designed to work well with pixel input, and it is unclear how it should be modified to work on the more challenging pixel input Adroit benchmark.
>
> As we discussed in our related work section, there are a large number of papers on offline RL, “...However, most of these works are tested on tasks with proprioceptive (positions & velocities) input, and they do not consider pretraining from non-RL data.” One question we might ask is, can we convert these offline algorithms to work with pixel-input? This is certainly possible, but it is not the focus of our research. In our case, in the end, we found DrQv2 + SafeQ + smaller encoder learning rate to be the best choice because DrQv2 is an extensively tested SOTA method on pixel-input tasks, and this combination has a minimalist design with superior performance and efficiency.
>
> The baselines we use for Adroit are, in fact, SOTA baselines. RRL and FERM are shown to work well on Adroit with pixel input (again, it is important to note that we are concerned with the more challenging **pixel-input** setting). And RRL is the **only method**, prior to our work, that achieved non-trivial performance on the hardest Relocate task. In fact, we made our FERM baseline **even stronger** by reimplementing it and using a new set of hyperparameters (discussed on page 5, section 4 “Getting stronger baselines”), giving a much stronger baseline performance compared to the original authors’ implementation. And we also made our DrQv2 baseline stronger for a fair comparison in DMC.

---

> ### Author Response · Authors · 2022-08-02
> **Response to Reviewer i5GH (1)**
>
> > Weaknesses: While the overall approach and results look promising, the current submission unfortunately lacks technical depth and rigor. In the introduction section, the authors outline their main contributions. Unfortunately, I do not believe there is sufficient evidence to rigorously substantiate these claims.
>
> > "Novel framework..." -- I believe there is limited novelty in this submission, especially in light of several recent papers (most of which are published by now). For example, the current submission does not cite or discuss MVP[1], PVR[2], R3M[3] which all explore training encoders from out of domain data.
>
> Thank you for providing these concurrent related works. Firstly, our work was done earlier than all the works you mentioned. Due to the double blind reviewing policy, we cannot show the evidence in this rebuttal, but we can prove this to AC if you find it necessary.
>
> Besides, we believe that our work has a sufficient amount of new contributions compared with these works. More details are discussed below. We will summarize these discussions/comparisons in the related work section of the final version.
>
> **MVP paper**: MVP uses self-supervised pretraining on images collected online and is tested on a new suite of control tasks proposed by the authors. The main differences are:
> - We consider the combination of **non-RL data, offline RL, and online RL data**. This paper only studies how non-RL data can be combined with online RL.
> - Our work can be seen as **a more general** framework in the sense that if we remove stage 2 training of VRL3, set encoder learning rate to 0, and switch to a different dataset and pretraining method in stage 1, then we arrive at the MVP framework.
> - **Novel results** found in our paper are not covered by the MVP paper: we provide a comprehensive study on the effect of enabling/disabling encoder training in different stages, and discuss how it can be finetuned in a stable and effective manner, we discuss how self-supervised learning in stage 2 might be entirely unnecessary when offline RL updates are used, etc.
> - One of our major contributions is a new SOTA performance on the popular and challenging Adroit benchmark. MVP **does not** study this benchmark. MVP also does not benchmark against prior SOTA algorithms in robotic control such as RRL and FERM (figure 5 of MVP paper).
>
> **PVR paper**: an interesting paper that combines pretrained encoders and imitation learning. The main differences are:
> - Our paper is focused on combine encoder pretraining with offline and online **RL**, while PVR studies how pretraining is combined with **imitation learning**.
> - Our work can be seen as **a more general** framework in the sense that if we disable stage 3 training of VRL3, set encoder learning rate to 0, enable BC training in stage 2, and switch to a different pretraining method in stage 1, then we arrive at the PVR framework.
> - **Novel results** found in our paper are not covered by the MVP paper: for example, we show that imitation learning (behavioral cloning) in stage 2 can be entirely removed when proper offline RL updates are applied, etc.
> - PVR has tested on the Adroit environment, they use more demonstrations than us (100 demos in PVR, while we use the standard 25 demos), and their performance is **lower than ours** (they achieve an average of 85% success rate, shown in Figure 1 of PVR paper, while we reach 95% or higher).
> - Pretraining excluded, PVR reports a wall-clock training time of 8-24 hours (appendix A.5 of PVR paper), while we report 1.6-13.3 hours (an average of 5.57 hours, see appendix B.3 (page 34) of our paper) training time to reach a stronger performance, which is **much faster**.
>
> **R3M paper**: similar to the PVR paper, they combine pretrained encoder (using a more sophisticated pretraining method) and imitation learning. The differences are similar to when we compare VRL3 and PVR. Performance-wise, R3M reports a <70% success rate on the Adroit benchmark, **weaker than** our 95% success rate. In terms of computation efficiency, it is not reported in the R3M paper, we wrote an email to the first author and are waiting for their reply. (So there is still a possibility that although they reach a weaker performance, they reach it with less wall-clock time.)
>
> In summary, these are very interesting papers, we have cited them and discussed them in our revision. However, these papers do not cover the unique contributions in our paper, and their performance on Adroit is weaker than our results, based on these results, it seems our work is indeed the first successful framework that achieves a new SOTA performance on pixel-input Adroit while utilizing a combination of non-RL, offline RL and online RL data.

---

> ### Author Response · Authors · 2022-08-07
> **Follow-up response to Reviewer i5GH (discussion phase)**
>
> We would like to first thank you again for your helpful comments and questions. Since we are near the end of the discussion phase, we would like to post a follow-up response.
>
> In our previous response and our revision, we have addressed your main concern regarding our related works. In summary:
> - Our work is done **prior to all these works**. If needed, evidence can be provided to PC to prove this point without compromising anonymity.
> - For the 3 papers you recommended, we read them carefully and pointed out exactly how their frameworks are different from ours, and none of them consider our setting. Note that just "explore training encoders from out of domain data" is nothing new, as discussed in our section 2.2 on page 3. This is **not** our main contribution, instead, our framework is novel in it **combines non-RL, offline RL, and online RL data**, which is not explored in prior (and even concurrent) works.
> - We also pointed out that **none of these 3 papers** show better performance on Adroit from pixels. So our claim on obtaining a SOTA performance on Adroit is in fact, accurate.
> - We further identified **another 17 papers** (many of which were released on Arixv in the past few months) and show that even if we consider these papers, the novelty of our framework and our SOTA performance are **still valid** (appendix D.5, page 40). And
> after incorporating reviewer suggestions, we found that our performance can be **further improved** in Relocate with a higher capacity encoder, leading to **24 times** better sample efficiency than RRL.
> - We also pointed out that AWAC is **not a SOTA on Adroit with pixel-input**. AWAC has been tested on Adroit with proprioceptive input, but not on the more challenging pixel-input setting. Additionally, our extensive experimental results demonstrate a number of important advantages of our design: it is sample efficient, computationally fast, and can achieve a new SOTA performance without using more sophisticated offline-online methods.
>
> It seems to us your main concern has been fully addressed. Do you find this response satisfactory? Or if you are aware of any other recent papers that we should cite and discuss, it would be great if you can post them during the discussion phase, so we can add them to our paper and submit a new revision before the discussion phase ends. Thank you very much for your time and effort in providing valuable feedback on our work!

---

### Author Response · Authors · 2022-08-03
**General response**

We thank all reviewers for your time and effort in providing these insightful feedback that help us improve our work. Our revision has been uploaded. The main text and the appendix are combined, and the modifications in the main text are highlighted in blue for a better reading experience.

We are encouraged by the positive feedback on multiple aspects of our work: our design is recognized as "straightforward" and "very practical". Our writing and presentation "a breeze to read". Our results are "promising", "impressive" and "much higher than previous baselines", and our experiments are "extensive", providing "valuable insight for the ML community" and "shed light on future researchers who is willing to jump into this field".

Here we summarize the major changes in our revision:

- **[i5GH]** After looking at the recommended papers and a careful search online, a total of 20 recent papers are discussed in Appendix D.5 (page 40) to show our framework is indeed novel and we achieve SOTA on Adroit with pixel input.
- **[5Wwb]** Comparison and discussion on 2 new BC baselines added to Appendix A.8 (page 30), showing naive value learning can hurt performance.
- **[i5GH, 5Wwb]** CCE ablations added to Appendix A.9 (page 31), showing that CCE is still the best fit for our framework, due to its simplicity, negligible overhead, and performance.
- **[5Wwb, vuXw, 1wyj]** Experiments with deeper and wider encoders added to Appendix A.7 (page 29), showing a higher capacity encoder can further improve performance on Relocate, solving it 24 times faster than RRL.
- **[1wyj]** Writing is refined. We emphasize our quantitative evidence, or provide a pointer to supporting figures and numbers, whenever possible. Emphasized our hyperparameter sensitivity study, which shows VRL3 is robust and only adds one additional hyperparameter that is important and requires tuning.
- **[1wyj]** Additional discussion on data augmentation added to Appendix D.4 (page 39).
- **[vuXw]** Comparison to long-term performance of RRL added to Appendix A.6 (page 28), showing that VRL3 at 4M is slightly stronger than RRL at 12M.
- Some other minor edits are made to improve the paper.

If you have other concerns or comments, please feel free to post them during discussion period. We are happy to answer any other question you may have.

---

### Comment · Area_Chair_tVhn · 2022-08-09
**Reviewers please check the authors responses.**

Dear reviewers, could you please check the authors' responses and see if they sway your overall assessments of this paper? Either way, we would greatly appreciate it if you acknowledged that you have read the responses (thanks to 5Wwb who have already done so) and discussed any follow-up questions with the authors.

---

### Meta-Review · Area_Chair_tVhn · 2022-08-25

**Recommendation:** Accept
**Confidence:** Less certain

**Metareview:**

This paper introduced a simple paradigm for improving sample efficiency of training deep reinforcement learning policies for vision-based control tasks. The idea is to use a 3-stage pipeline: 1) pre-training visual representations on large-scale image datasets, 2) policy training with offline RL, and 3) fine-tuning the policy with online RL. This work received mixed reviews from four reviewers, with one Reject, one Weak Reject, and two Weak Accepts. The reviewers appreciated the demonstrated effectiveness of the proposed approach despite the simplicity of the approach. Meanwhile, they expressed major concerns regarding the limited novelty concerning the burgeoning body of literature on visual pre-training and limited evaluations and ablation studies.

The authors drafted very detailed responses to the reviewers' comments, which clarified many technical issues brought up in the initial reviews. At the end of the discussion period, Reviewer i5GH (who did not engage in the discussions) and Reviewer 1wyj maintained their negative ratings of this paper, while the other two voted Weak Accept.

The AC read the paper, the reviews, and the authors' responses carefully. Reviewer i5GH's main criticisms are 1) the novelty of VRL3 and missing citations and discussions of prior work (MVP, PVR, R3M, etc.) and 2) insufficient comparisons and ablations of key model designs. The AC checked the publication/release dates of the mentioned works and believed they should be considered *concurrent* with this submission. Thus, the technical merit of this work should not be penalized by the existence of these related works. Meanwhile, the authors added the citations and discussions about these works in the revised draft, which addressed the reviewer's comment. In addition, the authors also provided additional ablation studies and clarifications which addressed the second point raised by Reviewer i5GH.

Reviewer 1wyj expressed concerns about the heavy revision during rebuttal and the overclaim and over-selling of the approach. The AC agreed with this reviewer that some language, such as "minimalist", should be toned down in the next reversion of this manuscript.

Taking all these into account, the AC found that the rebuttal has addressed the major issues raised in the reviews. Even though this work does not generate revolutionary ideas, it has shown convincing evidence of a practical approach that improves the learning efficiency of deep reinforcement learning in challenging vision-based control tasks. This work may pave the road for future work to develop more advanced methods. Therefore, the AC thinks that this work has passed the bar of acceptance at NeurIPS, despite the mixed final ratings.

**Award:**

No

---

### Decision · Program_Chairs · 2022-09-14

Accept